# Towards symmetry driven and nature inspired UV filter design

Michael D. Horbury[1,6]*, Emily L. Holt [1,2], Louis M.M. Mouterde [3], Patrick Balaguer[4], Juan Cebrián[5], Laurent Blasco[5], Florent Allais[3] & Vasilios G. Stavros[1]*

In plants, sinapate esters offer crucial protection from the deleterious effects of ultraviolet radiation exposure. These esters are a promising foundation for designing UV filters, particularly for the UVA region (400 – 315 nm), where adequate photoprotection is currently lacking. Whilst sinapate esters are highly photostable due to a *cis-trans* (and *vice versa*) photoisomerization, the *cis*-isomer can display increased genotoxicity; an alarming concern for current cinnamate ester-based human sunscreens. To eliminate this potentiality, here we synthesize a sinapate ester with equivalent *cis*- and *trans*-isomers. We investigate its photostability through innovative ultrafast spectroscopy on a skin mimic, thus modelling the as close to true environment of sunscreen formulas. These studies are complemented by assessing endocrine disruption activity and antioxidant potential. We contest, from our results, that symmetrically functionalized sinapate esters may show exceptional promise as nature-inspired UV filters in next generation sunscreen formulations.

[1] Department of Chemistry, University of Warwick, Gibbet Hill, Coventry CV4 7AL, UK. [2] Molecular Analytical Science Centre for Doctoral Training, Senate House, University of Warwick, Coventry CV4 7AL, UK. [3] URD Agro-Biotechnologies Industrielles (ABI), CEBB, AgroParisTech, 51110 Pomacle, France. [4] IRCM, Inserm, Univ Montpellier, ICM, Montpellier, France. [5] Lubrizol Advanced Materials, C/Isaac Peral 17-Pol. Industrial Cami Ral, 08850 Gava, Spain. [6] Present address: School of Electronic and Electrical Engineering, University of Leeds, Leeds, LS2 9JT, UK. *email: m.d.horbury@leeds.ac.uk; v.stavros@warwick.ac.uk

**N**aturally occurring sinapate esters[1] have shown promise as starting points for a generation of ultraviolet (UV) filters that offer exemplary photoprotection. They exhibit high levels of photostability under UV exposure, due to an efficient *trans-cis* and *cis-trans* photoisomerization resulting in a photo-equilibrium between these isomers. More specifically, the photoisomerization is proposed to consist of three dynamical processes in the excited electronic state. After the initial absorption of UV radiation, the sinapate esters undergo a geometry relaxation. The geometry relaxation is then followed by evolution on the excited state potential energy surface followed by photoisomerization, which forms either the *cis-* or *trans*-isomer, mediated by a conical intersection between the excited and ground electronic states[2–5]. During this relaxation, the excess electronic energy is converted into vibrational energy, leading to the isomers being formed vibrationally hot. This vibrational energy is eventually lost to the surrounding solvent bath[6,7]. These processes are illustrated in Fig. 1 (along with their associated rate constants); starting from the *trans*-isomer of the ester ethyl sinapate (ES) as an example.

Not only do these sinapate esters display high UV photostability, they also demonstrate potent antioxidant capabilities[8]. However, they are not without their issues, which must be taken into consideration if to be included in any future sunscreen formulation: firstly, their absorption does not completely span the UVA region (400–315 nm), thus lacking optimum UVA photoprotection; secondly, their UVA $\lambda_{max}$ is close to the UVB (315–280 nm), of which there are already a plethora of effective UVB filters; thirdly, the two isomers have differing absorption profiles, with the *cis*-isomer having (in general) the weaker absorption[4,5,9]; finally, the genotoxicity of the *cis*-isomer has been shown to be significantly higher in a related cinnamate[10]. One must also keep in mind that given the growing concern over several other EU and FDA approved UV filters flagged as human-toxic[10,11] and eco-toxic[12–14], this inevitably adds further considerations before any sunscreening agent can be included in a sunscreen formulation.

A highly intuitive solution to the issues introduced by the *cis*- and *trans*-isomer conundrum is to add identical ester moieties across the acrylic double bond, leading to indistinguishable *trans*- and *cis*-isomers. Concurrently, this serves to increase π-system conjugation and thus red-shift the UVA $\lambda_{max}$, when compared to the exemplar sinapate ester ES. Indeed, ES has been previously studied using femtosecond (fs) transient electronic (UV/visible) absorption spectroscopy (TEAS) by our group[4]. TEAS has proven to be a powerful tool for observing the photoisomerization of sinapate esters, particularly in identifying the formation of any photoproducts[2–5,7,15]. However, while the addition of identical ester moieties across the double bond spectrally shifts (to lower energy) in the absorbance (a positive attribute towards UVA filter design), this alone does not indicate whether such an approach maintains the desired properties (*vide supra*) that may facilitate symmetry driven sinapate esters being promising UV filters.

To address this, we have synthesized diethyl 2-(4-hydroxy-3,5-dimethoxybenzylidene)malonate (diethyl sinapate), abbreviated DES hereon. Structures of both DES and its precursor ES are shown in Fig. 1. In this instance, we utilize a combination of TEAS and steady-state spectroscopy to investigate the photostability of DES. The combination of time-resolved and steady-state spectroscopies, enables one to link the ultrafast with the ultraslow dynamics, providing detailed insight into how photophysical processes involved at the very early stages of the photon-molecule interaction, influence the longer-term photostability. TEAS measurements were taken of DES blended with a commercial sunscreen emollient, C12–15 alkyl-benzoate (AB) deposited on a synthetic skin mimic, VITRO-CORNEUM® (VC). The dynamical measurements taken using this innovative approach, makes the data accrued directly valid to real-world applications of sunscreen formulas, with potentially transformative repercussions to the cosmeceutical industry. We also make crucial advances in other aspects of the work through: (a) developing two greener synthetic procedures to produce DES; (b) performing endocrine disruption measurements of DES for the alpha oestrogen receptor (ERα) and the xenobiotic receptor (PXR); and (c) determining the antioxidant potential of DES using a 2,2-diphenyl-1-picrylhydrazyl (DPPH) assay. The spectroscopic advances along with (a)-(c) enable us to propose whether DES shows promise as a potential nature-inspired sunscreening agent.

## Results

**Synthesis of DES**. DES can be readily obtained through the Knoevenagel-Doebner condensation of syringaldehyde and ethyl malonate. All the synthetic procedures that have been reported in the literature for such a condensation are not only quite

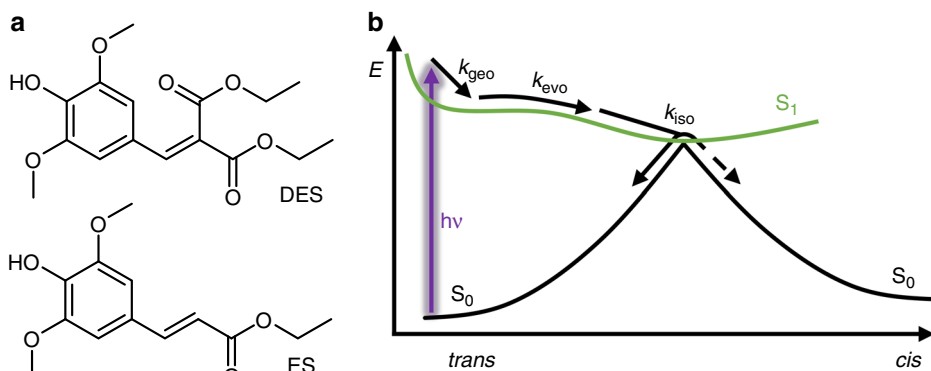

**Fig. 1** Sinapate structures and photoisomerization scheme. **a** The geometric structures of diethyl sinapate (DES) and ethyl sinapate (ES), demonstrating the symmetrical functionalization of DES compared to its precursor ES. **b** Schematic illustrating the dynamical processes involved in the photoisomerization of ES. Following photoexcitation with a UV photon (represented by the purple arrow), to the first electronically excited (S₁) state (green curve), *trans*-ES undergoes three processes during its subsequent relaxation to the electronic ground (S₀) state (black curve). Each process is indicated by a labelled arrow: $k_{geo}$ represents the rate of the geometry rearrangement; $k_{evo}$ is the rate of evolution in the electronic excited state, such as vibrational cooling, along with solvent rearrangement; and $k_{iso}$ is the rate for the photoisomerization. At the S₁/S₀ conical intersection, where the green and black curves meet, there are two possible pathways: relaxation to the S₀ state of the *trans*- or *cis*-ES species, represented by the solid black and dashed black arrows respectively

hazardous, as they use toxic reagents/solvents such as piperidine and benzene, but they also require energy consuming conventional heating. Here we use microwave-heating to reduce energy consumption and eliminate benzene, while reducing the reaction time from 7.5 h to 30 min. Although this procedure brought significant improvements, it still required piperidine; moreover, one could also question the relevancy of microwaves at the industrial scale. A proline-mediated Knoevenagel-Doebner condensation in ethanol[16] under conventional heating, recently developed in our group, was successfully implemented to the synthesis of DES at the multigram scale allowing for full replacement of piperidine. Finally, whatever the synthetic procedure used (i.e. microwave-assisted or proline-mediated condensation), we also succeeded in replacing column chromatography purification by a simple precipitation. Ultimately, this leads to a more sustainable and environmentally friendly synthetic route of DES.

**Transient absorption spectroscopy.** The transient absorption spectra (TAS) of DES in AB (10 mM) after being applied to the surface of VC (termed DES VC/AB hereon) and allowed to absorb into the substrate, are shown as a false colour map in Fig. 2a. TAS of DES in AB (1 mM) are also shown in Supplementary Fig. 1 for comparison. We note the difference in initial DES concentration. This is due to the dilution of the sample as it absorbs into VC, meaning we are unable to accurately determine the concentration of DES on VC. That being said, due to the signal strength in the TAS (in comparison to the TAS for DES in AB), we modestly estimate ~1.5 mM. Additional TAS were collected of DES in ethanol and cyclohexane (see Supplementary Figs. 2 and 3) to provide a range of solvent environments as comparisons. A saturated solution (<1 mM) of DES in cyclohexane was used, while in ethanol a concentration of 1 mM was used. All samples were photoexcited at their UVA $\lambda_{max}$: VC/AB = 335 nm; AB = 335 nm; ethanol = 336 nm; and cyclohexane = 325 nm (see Supplementary Fig. 4 for UV/visible spectra).

After initial excitation, likely due to an optically bright $^1\pi\pi^\star$ state, akin to what has been seen in ES[4], the TAS of DES VC/AB consists of a single excited state absorption which rapidly decays within a pump-probe time-delay ($\Delta t$) of <100 fs (not apparent in Fig. 2, see Supplementary Fig. 5) as the excited population on the

$^1\pi\pi^\star$ state evolves from the Franck-Condon region. This has been previously assigned to geometry relaxation (intramolecular vibrational redistribution)[17,18], labelled $k_{geo}$ in Fig. 1, and we conjecture a similar mechanism is in operation here. The geometry relaxation reveals three distinct spectral features that consist of: (i) a ground state bleach (~350 nm) corresponding to where the DES electronic ground state absorbs; (ii) a strong excited state absorption (~380 nm); and (iii) a second weaker excited state absorption (~540 nm). (i)–(iii) are in accordance with what has been observed from $^1\pi\pi^\star$ state-driven dynamics for ES[4], indicating that the additional $CO_2Et$ has little influence in the spectroscopic signatures of DES relative to ES; we return to discuss this further in the post narrative.

As the TAS evolve in time, the excited state absorption at ~540 nm has decayed to zero $\Delta OD$, (where $\Delta OD$ denotes change in optical density) by $\Delta t = $~4 ps. The decay of the excited state absorption (both features) is attributed to, in part, repopulation of the electronic ground state from the $^1\pi\pi^\star$ state. This is likely mediated by a photoisomerization pathway, labelled $k_{iso}$ in Fig. 1[4]. However, a remnant of the excited state absorption at ~380 nm and the ground state bleach feature remain. In previous spectroscopy-driven studies on sinapates, this has been attributed to the formation of the isomer photoproduct, or due to a phenoxyl radical species due to an instantaneous two-photon ionization[2,4,6,7,19,20]. However, in DES the cis- and trans-isomer are identical (ruling out the isomer photoproduct) and both the absorption at 380 nm and the ground state bleach decay over time. The decay of these features rules out the formation of the radical as it would be expected to persist beyond the maximum $\Delta t$[2,4,6,7,19,20]. This alludes to additional transient species being involved in the relaxation of DES, not seen in ES or any related sinapate or cinnamate in the condensed phase[2,4,5,7,15,19]. Whilst, it appears that these features decay completely back to the baseline, closer examination of the TAS at $\Delta t = 2$ ns, shown in Fig. 2c, shows that a very small amount of the ground state bleach is still present, which is not attributed to the solvent; see Supplementary Fig. 7 for solvent-only TAS. We add that we do not see evidence of vibrational cooling of the electronic ground state, as previously seen for ES[4]. This is due to the presence of the absorption feature at 380 nm masking the spectral signature

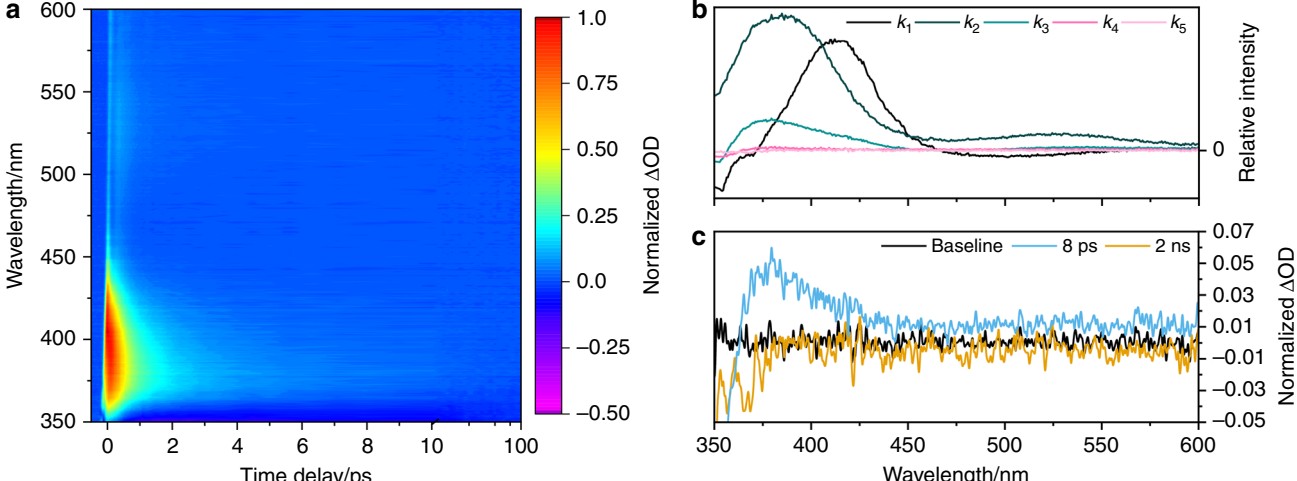

**Fig. 2** Ultrafast spectroscopy results for DES. **a** Transient absorption spectrum (TAS) of DES in C12–15 alkyl benzoate deposited on a synthetic skin mimic, VITRO-CORNEUM® (VC/AB) photoexcited at 335 nm, shown as a false colour map, with the intensity scale representing a change in normalized optical density ($\Delta OD$). The time-delay is plotted linearly from −0.5 to 10 ps then as a log scale from 10 to 100 ps. **b** Evolution associated difference spectra (EADS) from the sequential global fit of the TAS of DES in VC/AB photoexcited at 335 nm. Zoomed-in plots showing only $k_4$ and $k_5$ for this system can be found in Supplementary Fig. 6a). **c** Selected TAS at specific $\Delta t$ highlighting the absorption at 380 nm (8 ps, green) and incomplete ground state bleach recovery (2 ns, orange)

associated with vibrational cooling. The absence of vibrational cooling has been seen in several other sinapates and cinnamates[2,3,21].

To recover the kinetic parameters from the TAS presented (see Fig. 2a and Supplementary Figs. 1–3), we carried out a sequential $(A \xrightarrow{k_1} B \dots \xrightarrow{k_n} n)$ global fit, across the entire spectral region of our probe, using the software package Glotaran[22,23]. The rate-constants ($k_n$) for DES in VC/AB returned from the sequential global fit are shown in Table 1 (the corresponding rate-constants for DES in other solvents are shown in Supplementary Table 1), while the evolution associated difference spectra (EADS) are shown in Fig. 2b and Supplementary Figs. 1–3. Additional zoomed-in plots of EADS associated with $k_4$ and $k_5$ are also shown in Supplementary Fig. 6. We also add that solvent only time-zero responses (representing our instrument response) are shown in Supplementary Fig. 8 and the residuals to all fits are shown in Supplementary Fig. 9. Furthermore, the EADS of $k_4$ and $k_5$ are overlaid with the corresponding TAS at a comparable time delay; these are shown in Supplementary Fig. 10. Finally, we note that a more concentrated solution of DES in ethanol (10 mM, the highest possible within our experimental constraints) was tested for evidence of aggregation. There was no evidence (by comparison with the 1 mM counterpart) to suggest that DES aggregates were formed. Further studies would be warranted at higher concentrations, but this is beyond the scope of current experimental capabilities.

**Steady-state spectroscopy.** To complement the time-resolved measurements, steady-state irradiation measurements were carried out, to determine the long-term photostability of DES.

**Table 1 Summary of rate constants**

|  | $k_1/s^{-1}$ $(\times 10^{12})$ | $k_2/s^{-1}$ $(\times 10^{12})$ | $k_3/s^{-1}$ $(\times 10^{11})$ | $k_4/s^{-1}$ $(\times 10^{10})$ | $k_5/s^{-1}$ $(\times 10^{8})$ |
|---|---|---|---|---|---|
| VC/AB | 7 ± 2 | 3.0 ± 0.3 | 4.24 ± 0.07 | 1.02 ± 0.06 | « 5 |

Rate-constants ($k_n$) resulting from the sequential global fit of the TAS of DES in VC/AB shown in Fig. 2a. The errors are quoted to 2σ. Rate-constants for DES in AB, ethanol and cyclohexane can be found in Supplementary Table 1

UV/visible spectra were taken at various time-intervals during the irradiation of the sample at its UVA $\lambda_{max}$ at solar intensity (0.2 mW cm$^{-2}$). The resulting UV/visible spectra of DES in AB are shown in Fig. 3. It is clear from these spectra that over a period of two hours, DES only experiences a minor reduction in its absorbance, 3.3%, while for the UVA $\lambda_{max}$ of ethanol and cyclohexane a drop of 3.1% and 1.6%, respectively, was observed (see Supplementary Fig. 11 for additional spectra). Hampered by scattering issues within the spectrometer, we were unable to perform these measurements for DES in VC/AB. For comparison, *trans*-ES in cyclohexane experiences a 16% loss in absorbance over a period of 45 min, a consequence of establishing a photo-equilibrium between the two (*trans* and *cis*) structural isomers[4].

In addition to the photostability measurements of DES, we have also calculated the critical wavelength of DES from its UV/visible spectrum in ethanol, see Supplementary Methods for additional methodological information. Critical wavelength is the industrial standard for determining if there is sufficient UVA

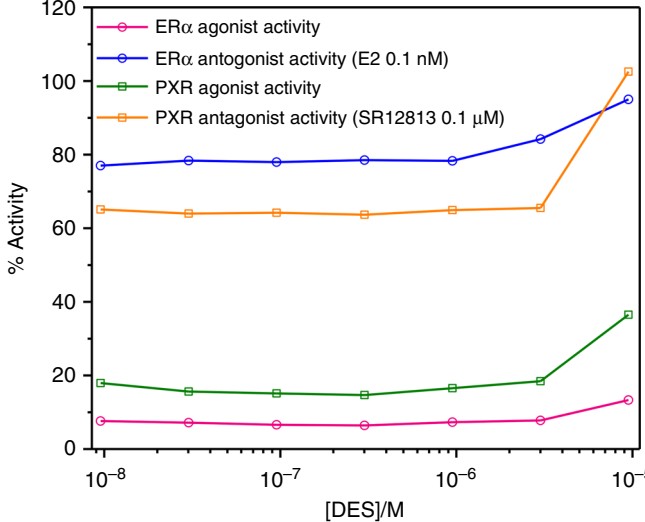

**Fig. 4** Endocrine disruption assay results. Endocrine disruption activity of DES, as determined by assays on several cell lines, for the alpha oestrogen receptor ERα (circle) as either an agonist (magenta) or antagonist (blue), and for the xenobiotic receptor PXR (square) as either an agonist (green) or antagonist (orange). The activity of ERα is regulated by the steroid and oestrogen sex hormone 17β-estradiol (E2). A known agonist for PXR is SR 12813

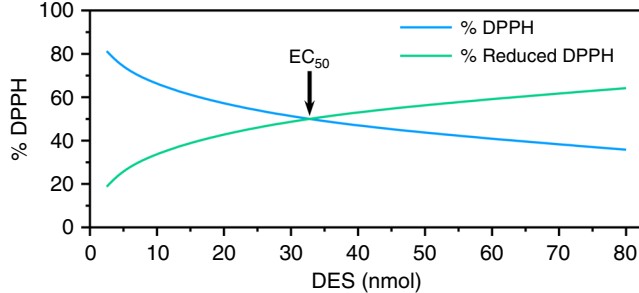

**Fig. 5** Determination of the antiradical activity of DES. The determination of the antiradical activity of DES has been determined via 2,2-diphenyl-1-picrylhydrazyl (DPPH) assay. These tests involve adding DES solution in ethanol at different concentration to homogeneous DPPH solution. The $EC_{50}$, i.e. the amount of DES needed to reduce the initial number of DPPH free radicals by half, is provided by the crossing point of % DPPH (blue) and % reduced DPPH (green), which occurs at 32.7 nmol

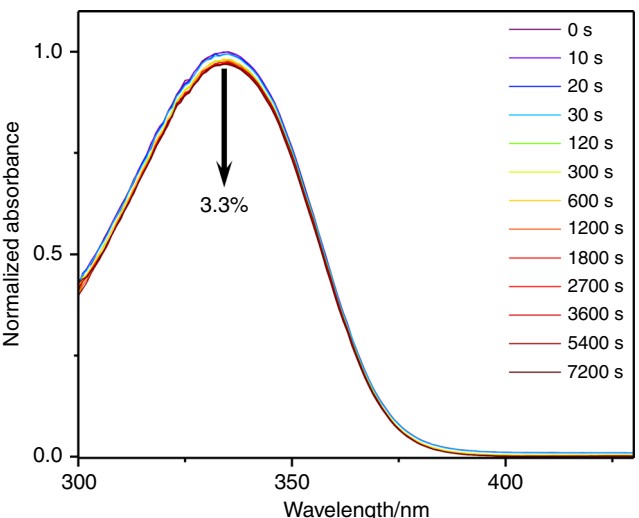

**Fig. 3** Long-term photostability of DES. UV/visible spectra of DES in C12–15 alkyl benzoate (AB), at varying durations of irradiation at 335 nm and replicating solar intensity. The downwards arrow (in black) denotes the observed 3.3% decrease in absorbance after 7200 s of irradiation

| Table 2 Antioxidant potentials | | | | | | |
|---|---|---|---|---|---|---|
| | Irganox 1010 | Trolox | BHT | BHA | ES | DES |
| EC$_{50}$ (nmol) | 6.9 | 4.0 | 7.1 | 3.7 | 13.7 | 32.7 |
| [antioxidant]/[DPPH] | 0.18 | 0.11 | 0.19 | 0.10 | 0.36 | 0.86 |

EC$_{50}$ (nmol) and [antioxidant]/[DPPH] values for DES, ES and several commercially available antioxidants

protection. The value we retrieved was 364 nm (see Supplementary Fig. 12).

**Endocrine disruption potential of DES**. Endocrine disruption activity of DES was assayed on two types of receptors, see Supplementary Methods for details. The first receptor selected was the ERα receptor, which is a member of the nuclear hormone receptors family, whose activity is regulated by the steroid and oestrogen sex hormone 17β-estradiol (E2). The second receptor is the PXR receptor, a member of the steroid and xenobiotic sensing nuclear receptors family, with a known agonist being SR 12813. Endocrine disruption assays, shown in Fig. 4, demonstrate that DES is neither an agonist nor antagonist ligand for either ERα or PXR.

**Antioxidant potential of DES**. While the photostability of sinapates is a major interest for use as UV filters, their potent antioxidant capabilities are an added benefit. Therefore, to determine if DES also exhibits antioxidant potential, a DPPH assay was carried out. This method determines the H-donor capacity of the antioxidant to quench the stable DPPH free radical, as previously reported[8]. In this study, the EC$_{50}$ value corresponds to the amount of antioxidant needed to reduce half of the initial population of DPPH radicals. The lower the EC$_{50}$ value, the higher the antioxidant potential. DES's EC$_{50}$ value was determined as being 32.7 nmol, as highlighted in Fig. 5. For ease of comparison with other studies, we have converted this value to a ratio, quoted as [antioxidant]/[DPPH]; this gives a value of 0.86 for DES. In addition, we have included DPPH assays on several commercially available antioxidants, presented in Table 2.

**Discussion**
The spectroscopic measurements, both time-resolved and steady-state, have demonstrated that the addition of the second ester moiety has not impeded the high UV photostability that the sinapate backbone possesses[4]. Indeed this has had the opposite effect, highlighted by the minimal drop (≤3.3%) in the absorbance of DES after 2 h of UV irradiation at solar intensities, see Fig. 3. Alongside this increase in photostability, the critical wavelength of DES has significantly red-shifted compared to ES, cf. 364 nm for DES compared to 352 nm for ES. While this falls slightly short of current UVA filters, i.e. avobenzone is 378 nm, it is clearly a step in the right direction (see Supplementary Fig. 12 for details). We shall now focus on the photochemistry responsible for the apparent DES photostability.

To recover the photodynamics observed in the TAS, we performed a sequential global fit across the entire spectral window of our probe. This resulted in four dynamical processes, described by rate constants $k_{1,2,3,4}$, being recovered; we note that $k_5$ is used to describe the long-lived ground state bleach, which does not recover within the time-window of our experiment (except for the cyclohexane data, in which all spectral features return to baseline). We recognized that there are multiple sources we can attribute this incomplete ground state bleach recovery including a potential molecular photoproduct, or a trapped excited state population such as a triplet state (note an absorption feature and

no ground state bleach is observed for DES in ethanol, see Supplementary Fig. 2). Triplet states have been seen previously for sinapate and cinnamate esters in the gas-phase, with their population being mediated by an nπ* state[7,24,25]. These could be a potential source of the small depletion in our steady-state irradiation studies (see Fig. 3 and Supplementary Fig. 11), qualitatively in line with the (very) minor contribution of EADS for $k_5$ (see Fig. 2b and Supplementary Fig. 6). From inspection of the EADS linked to $k_{1,2,3}$ (except for DES in cyclohexane where $k_2$ is absent) these appear to have similar features to EADS seen in related sinapate esters in polar solvents[3,4]. Therefore, we believe that the EADS for DES describe similar dynamical processes, whereby: $k_1$ represents a prompt geometry relaxation out of the Franck-Condon region; $k_2$ describes further evolution along the excited state, including vibrational cooling and solvent rearrangement; and $k_3$ is the rate-constant for photoisomerization, repopulating the electronic ground state. Under this description $k_1 = k_{geo}$, $k_2 = k_{evo}$ and $k_3 = k_{iso}$ (see Fig. 1). Importantly, we note that due to the symmetrical nature of DES, it is not possible to determine if the photoisomerization occurs completely or is an aborted photoisomerization[26].

Interestingly however, additional spectral and dynamical (through the returned $k_4$) features are observed in the TEAS measurements of DES when compared to related sinapate esters (e.g. ES) and cinnamates[2,4,5,7,15,19]. These features pertain to the ground state bleach and an excited state absorption at 380 nm. We believe that these features are likely due to an electronic excited state, the front-runner being most likely a $^1$nπ* state, previously implicated to play a role in the photodynamics of cinnamates[24,25,27,28]. The decay of the absorption at 380 nm corresponds to an almost complete recovery of the ground state bleach. Importantly, whether this additional decay pathway happens following bifurcation of the excited state population or subsequent to photoisomerization is unknown.

While the overall picture of DES photochemistry, mainly the photoisomerization (or aborted photoisomerization) is similar in all solvents and constitutes the main finding (from a dynamics viewpoint) of the present work, differences in these dynamics, reflected in the associated rate-constants (see Table 1 and Supplementary Table 1), do exist and warrant discussion. We also choose to focus our discussion on the differences between DES in VC/AB compared to DES in various solvent environments and, where appropriate, the insights we draw into the intrinsic properties of photoexcited DES when mounted on a skin mimic.

First, comparing DES in VC/AB and DES in AB, the major difference is the almost three-fold increase in $k_4$ for DES in AB compared to DES in VC/AB. The reason for this could rest in population trapped in this excited state, experiencing a greater barrier towards ground state recovery for DES in VC/AB. Second, comparing DES in VC/AB with DES in cyclohexane, the six-fold increase in $k_4$ for DES in cyclohexane may also be reconciled by relative barrier heights. Interestingly for DES in cyclohexane, there is complete ground state bleach recovery. Understandably, a decreased residence time in this state for DES in cyclohexane could also explain the apparent ground state bleach recovery, given there is less opportunity for competing pathways. The (positive) knock-on effects of this could (tentatively)

explain our steady-state irradiation data of DES in cyclohexane, which show the smallest amount of depletion following prolonged irradiation.

Like DES in cyclohexane, the TAS at 2 ns for DES in ethanol has no apparent ground state bleach, however the presence of a new absorption feature at ~350 nm is likely the cause for the absence of the ground state bleach. This peak appears to grow in as the absorption at 380 nm and the ground state bleach recover. We believe that this feature is due to the presence of the phenoxyl radical generated via an instantaneous two-photon ionization, as seen in numerous previous studies in related cinnamates and sinapates[2,20,21]. However, this absorption feature in the TAS is very small, and hence we are unable to confirm its two-photon (pump) dependency through TAS. We add here that the presence of the phenoxyl radical in the present measurements is an artefact of the ultrafast spectroscopic measurements; its two-photon dependence makes it highly unlikely to occur in nature. Unfortunately, the presence of this peak has also hindered our ability to accurately fit the TAS using the sequential global fitting model, thus we were only able to accurately extract $k_1$, $k_2$ and $k_3$, as the fit significantly overestimates $k_4$ for the decay of the absorption at 380 nm and ground state bleach recovery.

Ultimately, while this additional decay pathway is present, it does not appear to impact the long-term photostability of DES. Characterisation of this additional state and how it is populated will inevitably benefit from high level theoretical and (complementary) experimental work, the latter utilizing different probe techniques, to elucidate how the excited state population evolves in DES. This, however, is beyond the scope of the current work.

Aside from the photostability of DES, another consideration in its candidacy as a next generation sunscreen is whether or not it acts an endocrine disruptor, a topic which continues to draw controversies for current UV sunscreens[11–13]. Due to these controversies and the general concern over endocrine disrupting chemicals[29], it is becoming a significant factor in the design of sunscreen agents. The endocrine disruption measurements of DES showed no adverse effects to the ERα or PXR receptors.

Likewise, potent antioxidant potential is an additional benefit in sunscreen design. Whilst the DPPH assays demonstrate that DES can act as an antioxidant, its activity (0.86) is lower than both ES (0.36)[8], as well as antioxidants already used in commercial sunscreen formulas BHT (0.19), BHA (0.10) and α-tocopherol (0.21)[30]. These antioxidants are only included in sunscreening formulas in small quantities compared to UV filters. Therefore, while the antioxidant potential of DES is lower, its concentration will be significantly greater, thus alleviating its low antioxidant activity. We have also included Irganox 1010 (0.18) and Trolox (0.11), which are used in the polymer and pharmaceutical industry respectively.

Amidst growing concerns of increasing exposure of society to solar radiation, the results presented herein demonstrate the promising potential of a symmetry driven and nature-inspired sunscreen for commercial use in sunscreen formulations; particularly given the improved environmentally friendly synthetic route. The symmetric functionalisation across the acrylic double bond ensures that the cis- and trans-isomers are equivalent, negating concerns over genotoxicity of isomeric photoproducts. Concurrently, the absorption has been spectrally shifted into the UVA region, where there is a growing need for UV filters. Moreover, the overall photodynamics measured for DES in an emollient used in commercial sunscreen formulas are consistent when deposited on a synthetic skin mimic. It thus demonstrates that whilst the dynamics are mildly dependent on DES environment, it highlights the need of 'as close to a true environment' real-world setting for these measurements.

Our studies may thus provide a blueprint for tuning molecular functionality, ultimately aiding additional beneficial properties such as enhanced antioxidant potential by modifying the ester group, which has little impact on the photodynamics (cf. ES versus DES)[2–4,7]. It is worth noting that the ester groups may not necessarily need to be symmetrical per se. Asymmetrical substitution, provided the intrinsic properties of the cis- and trans-isomers are commensurate, i.e. photophysical and photochemical properties, antioxidant capacity and endocrine disruption activity, could be promising sources of additional UV filters. However, such an asymmetric substitution may require more sophisticated (i.e. expensive) synthetic procedures, as well as additional considerations when testing for suitability. Our demonstration of customized photoprotection has also much broader ramifications; not only can it be applied to materials for human photoprotection, but also towards developing materials for plastics and resins, given their exposure to UV radiation. Finally, DES displays no endocrine disruption activity, which is a significant requirement in next generation UV filters given ever-growing concerns over current filters on both human health and aquatic life[11–14].

## Methods

**Spectroscopy.** The TEAS setup used to observe the photochemistry of DES has been previously characterized by Greenough et al.[31,32], and is reproduced in detail here. Fundamental femtosecond laser pulses (3 W, 1 kHz repetition rate) with a central wavelength of 800 nm were derived from a Ti:Sapphire regenerative amplifier (Spitfire XP, Spectra-Physics), seeded by a Ti:Sapphire oscillator (Tsunami, Spectra-Physics). This fundamental pulse train was split into three 1 W beams, two of which were utilized for our TEAS experiments. The first 1 W beam was used to generate pump pulses with a fluence of 200–800 μJ cm$^{-2}$ using an optical parametric amplifier (TOPAS-C, Spectra-Physics). The second 1 W beam was split further into two portions: (i) 950 mW and (ii) 50 mW. Beam (i) can be used for harmonic generation, however this capability was not implemented here. The probe pulses were a broadband white light supercontinuum generated by focussing (ii) in a vertically translated CaF$_2$ window, providing a probe spectral window of 345–735 nm. The pump-probe time delay (Δ$t$) was varied by adjusting the optical delay of the probe pulse, the maximum obtainable Δ$t$ was 2 ns. The probe beam is collected after passing through the sample, by a pair of 2" diameter, UV-enhanced aluminium, off-axis-parabolic mirrors with a focal length of 100 mm; these mirrors were incorporated into the setup to compensate for the additional scatter induced by VC. The beam then passes through a CaF$_2$ lens to collimate the probe, before being focussed into a fibre coupled spectrometer (Avantes, AvaSpec-ULS1650F) by a second CaF$_2$ lens. Changes in the optical density (ΔOD) of the samples were calculated from transmitted probe intensities. Samples of DES were made to a concentration of 1 mM in C12–15 alkyl benzoate (Lubrizol), ethanol (absolute, VWR) and cyclohexane (99.99%, VWR). It should be noted that we attained a saturated cyclohexane solution. For DES on VITRO-CORNEUM® (IMS Inc., VC) a DES sample at a concentration of 10 mM in C12–15 alkyl benzoate was applied to VITRO-CORNEUM®. A higher concentration (10 mM) of DES in ethanol was used to test for evidence of aggregation. The sample delivery system was a flow-through cell (Demountable Liquid Cell by Harrick Scientific Products Inc.) consisting of two CaF$_2$ windows; the windows were spaced 100 μm apart. The sample was circulated using a diaphragm pump (SIMDOS, KNF) and replenished from a 25 mL reservoir to provide each pulse-pair with fresh sample. For VC/AB the sample thickness was measured to be 120 μm and was mounted on the front of a CaF$_2$ window.

Steady-state UV/visible absorption spectra of DES in AB, ethanol and cyclohexane, were using a UV/visible spectrometer (Cary 60, Agilent Technologies). The samples, of concentration ~25 μM, were irradiated with an arc lamp (Fluorolog 3, Horiba) for 2 h, with the UV/visible spectra taken at various time points, at the corresponding TEAS excitation wavelength, using an 8 nm bandwidth of the irradiation source. The fluence was set to 100–200 μJ cm$^{-2}$ to mimic solar incidence conditions.

**Synthesis of DES.** DES was synthesized in one step using either a microwave-assisted or a proline-mediated Knoevenagel condensation between lignin-derived syringaldehyde and diethyl malonate (Fig. 6)[33].

In the first synthetic procedure, syringaldehyde (4 mmol, 728 mg) and diethyl malonate (13 mmol, 2 mL) were mixed together. Piperidine (2 mmol, 200 μL) was then added to the reaction mixture, the tube sealed and placed into a Monowave 400 microwave system. Constant power (50 W) was applied until reaching a temperature of 100 °C which was then maintained for 30 additional minutes.

The reaction mixture was purified by either, Purification 1: flash chromatography (cyclohexane/ethyl acetate 8/2). Fraction containing the wanted product was evaporated under vacuum to give pure DES (1.1 g, 85%); or

**Fig. 6** Preparation of DES. DES synthesized through Knoevenagel condensation of syringaldehyde and diethyl malonate

Purification 2: The reaction mixture was cooled down to room temperature and added dropwise to a 1 N HCl aqueous solution (50 mL) at 0 °C. The resulting precipitate was recovered by filtration and washed with cold water resulting in pure DES (1.05 g, 80%).

In the second synthetic procedure, syringaldehyde (4 mmol, 728 mg) and diethyl malonate (13 mmol, 2 mL) were mixed together in ethanol (0.5 M, 8 mL). Proline (2 mmol, 235 mg) was then added and the reaction mixture was refluxed overnight. The reaction mixture was cooled down to room temperature and added dropwise to a 1 N HCl aqueous solution (50 mL) at 0 °C. The resulting precipitate was recovered by filtration and washed with cold water to afford pure DES (1.04 g, 80%).

**Antioxidant measurements**. 190 μL of homogeneous DPPH solution (200 μM) in ethanol was added to a well containing 10 μL of the potential antiradical molecule solution in ethanol at different concentrations (from 400 μM to 12.5 μM). The reaction was monitored by a microplate Multiskan FC, performing 1 scan every 5 min for 7.5 h at 515 nm. The use of different amounts of DES give the $EC_{50}$ value, which is described as the efficient concentration needed to reduce the initial population of DPPH by half.

This procedure has been applied to commercially available antioxidants to provide benchmark values: Irganox1010 antioxidant used in polymers, Trolox antioxidant used in the pharmaceutical industry, BHT and BHA antioxidants are used in the cosmetic and food/feed industries.

### Data availability

The TEAS (Fig. 2, Supplementary Figs. 1–3) and corresponding fitting residuals (Supplementary Fig. 9), UV/visible absorption spectroscopy (Fig. 3, Supplementary Figs. 4, 11, 12), DPPH assay (Fig. 4) and endocrine disruption data (Fig. 5), is freely available in the Zenodo data repository with the identifier https://doi.org/10.5281/zenodo.327515.

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

## Acknowledgements

The authors would like to thank the Warwick Centre for Ultrafast Spectroscopy (WCUS) for the use of the Cary 60 and Fluorolog 3. M.D.H. thanks the Leverhulme Trust for postdoctoral funding. E.L.H. thanks the EPSRC for a PhD studentship through the EPSRC Centre for Doctoral Training in Molecular Analytical Science, grant number EP/L015307/1. L.M.M.M. and F.A. thank the ANR for the SINAPUV grant (ANR-17-CE07-0046), and the Grand Reims, the Conseil Départemental de la Marne and the Region Grand Est for financial support. Finally, V.G.S. thanks the EPSRC for an equipment grant (EP/J007153), the Leverhulme Trust for a research grant (RPG-2016-055) and the Royal Society and Leverhulme Trust for a Royal Society Leverhulme Trust Senior Research Fellowship.

## Author contributions

M.D.H. and E.L.H. acquired and analysed the time-resolved and steady-state spectroscopic data (equal contributions) and prepared the manuscript. L.M.M.M., P.B. and F.A. conceived and performed the synthesis, conducted DPPH assay and endocrine disruption measurements as well as contributing to the preparation of the manuscript. J.C. and L.B. provided invaluable direction for modelling a closer-to-realistic sunscreen environment (synthetic skin mimic) as well as critiquing the manuscript. V.G.S. conceived the experiments and provided guidance in data analysis and interpretation, and the writing of the manuscript.

## Competing interests

The authors declare no competing interests.
