## [Peer Review File · Nature Communications]

Reviewers' comments:

Reviewer #1 (Remarks to the Author):

The paper by Horbury et al., reports the synthesis and evaluation of a new candidate for sunscreen formulations. The photophysical properties of the new compound together with its endocrine disruption potential and antioxidant activity are presented. Overall, the paper has undoubted merits as it tries to provide a comprehensive view of the behavior of the system. However, some parts of the manuscript lack the required level of detail. Clearly, the paper is not presented as a complete study and a more detailed analysis would be needed before the real potential of this molecule could be understood. Nevertheless, some points in the manuscript should be improved before publication.

-The symmetrical nature of the DES is invoked several times (including the paper title) as responsible for some of the improved properties with respect to ES. Certainly, this is a clever design and the point may well be true. The lack of photoproducts (as the photoisomerization leads to exactly the same compound) is definitively interesting. However, it is not clear if the symmetry along the central double bond is required for this. For instance, an asymmetric compound with, for instance, two different ester moieties, should probably have the same photophysical properties (as it happens with ES and DES). For two similar ester moieties, the UV absorption of both isomers should be very similar. In addition, as the cis isomer of ES is genotoxic, this hypothetical asymmetric compound could also provide negative endocrine disruption activity, as it happens with DES. Thus, is it really necessary to have a symmetric compound? If not, DES could be just an example of a set of compounds, based on ES, with improved features in terms of toxicity (but very dependent on substitution) and, maybe, in terms of photostability.

-Figure 1 includes a very schematic diagram of the processes involved in the photoisomerization of ES. Based on the results included in the paper, the case of DES is more complex: the additional dark excited state species suggested imply the presence of, at least, two different excited states. As stated by the authors, a deep study of the photochemistry of the compound is out of the scope of the paper. I agree with this statement. However, the lack of more detailed information on the mechanism makes risky to draw the conclusions offered by the authors.

The manuscript is definitively interesting and some of the results are relevant enough to be published in Nature Communications. However, I feel that the present version is too preliminary and the above-mentioned issues should be improved before publication.

Some additional suggestions:

-The synthesis of DES is quite straightforward. The claims that previous preparations use toxic compounds and require energy-consuming heating are strictly true, but probably imply unneeded

overselling. In this preparation, piperidine (toxic, according to the authors) is also used. The use of microwave and column chromatography is not industrially efficient.

-Different solvents and different concentrations are used throughout the paper. This makes difficult to compare the data from one experiment to another. For the most concentrated experiment (TAS in AB, Figure 2, 10 mM), this should result in a 3.2% solution which in turn would be far lower in a real formulation (too low for a significant photoprotection, probably).

-The discussion on the effects of solvent polarity is confusing. The lack of influence of the viscosity on k_2 (k_{iso}) is linked by the authors to a low amplitude movement (hula-twist?). However, data available could also be consistent with an aborted photoisomerization (and, thus, with a small nuclear movement).

Reviewer #2 (Remarks to the Author):

The authors of this manuscript presented transient absorption data on a diethyl sinapate (denoted DES herein), a sinapate ester derivative, as well as other data pertaining to the feasibility of utilising this molecule as an active ingredient in commercial sunscreen such as long-term photostability, endocrine activity and the added benefit of antioxidant activity. Adding an extra ester to make indistinguishable trans- and cis-form (in order to solve the toxicity issue of the cis-form) and to red-shift the absorption spectrum to cover more UVA is clever, and transient absorption experiments performed on VC, a stratum corneum mimic, is novel. However, the authors did not sufficiently motivate how the transient absorption data, the major result of this manuscript which much of the discussion is centred on, helped establishing that DES is a suitable candidate for sunscreen. Furthermore, the authors' interpretation of the transient absorption data is not as convincing as one would hope to see in a publication like Nature Communications (see below). However, given the importance of the subject which will be of interest to a broad range of readers, we recommend a major revision which should address the major and minor concerns listed below, followed by a future review.

Major comments:

1. The subtle differences between the transient absorption data of DES in AB/VC (Figure 2) and those in solutions (AB, ethanol and cyclohexane; Figures S2 – S4), e.g., that the very long-lived state is only observed in the former, and that the intensity of the stimulated emission is different in different environment, are interesting. We recommend that the authors centred the discussion on these differences, which will help improve the novelty of this work. Specifically, what significant insight can experiment in VC offer, compared to the standard experiments performed in solutions?

2. Similarly, the authors should comment on what information transient absorption can provide while other techniques cannot.
3. The description and discussion of the transient absorption results (text in the bottom of the right column of page 2 and the top of the left column of page 3) is confusing. The authors stated that the excited state absorption and the ground state bleach “decay at different rates” (once here and once in the beginning of Discussion), but as the reviewer understands, the fastest three decay components ($k_1 - k_3$) are identical between the two features, an additional long-lived component (k_4) is required to fit the ground state bleach trace. Furthermore, the fact that four decay components were observed in the ground state bleach would argue against a sequential decay mechanism, which was used by the authors to model their transient absorption data. Although the multiexponential nature of the ground state bleach recovery does not totally rule out sequential decay, especially in the presence of overlapping absorption signals, the author should at least comment on the possibility of a parallel mechanism.
4. It is a strange approach where the authors exclude the negative transient absorption signal due to ground state bleach in their global analysis. There is no reason why global fitting would fail when adding an additional decay component (from three to four exponentials).
5. The authors stated that the very long-lived excited state observed in AB/VC “is most likely an $n\pi^*$ state”, but this assignment is not substantiated. This state, with decay rate constant of k_4 , showed large variation in decay rate in different environment with different viscosities (Table S1). Based on this observation, the authors later stated that it “has large amplitude nuclear motion involved during the dynamics” (top of the right column on page 4). Are these two assignments mutually exclusive?
6. Related to comment #5 above: the authors tried to draw attention to that the decay rates change as the solvent viscosity changes, but the dielectric constants of the solvents are also different. Can the observed difference in rate due to dielectric environment instead of viscosity?

Minor comments:

1. Page 2, right column, first paragraph below Figure 2 caption: it is very difficult to see those three features from Figure 2a.
2. Page 2, Figure 2 (and Figures S2 – S4): it is very difficult to compare the decay traces in panels b and c because they are plotted on different time scales.
3. Page 3, the first line of the left column, it is unclear how the authors rule out radical.
4. Page 3, Figure 3 and associated text, the authors drew attention to the fact that a 3.3% decrease of DES is observed after two hours of UVA irradiation, and compare to the easily degradable trans-ES. However, how is this number compared to the common sunscreen molecules that are used in commercial sunscreen? Although the data seem to indicate that a small fraction of DES is degraded, but is there any harmful photoproduct formed (e.g., degrade to ES which the cis form is toxic)? Is this something that the authors can characterise?

5. Why do the authors believe that DES is the “next generation of sunscreen”? What is the advantage of DES compared to the current generation molecules and their blends? For example, what is the advantage of DES compared to zinc oxide which physically blocks UVA?

Reviewer #3 (Remarks to the Author):

Dear corresponding Author,

The manuscript was adequately written, containing original data regarding the potential use of DES as an UV filter.

Considering the high pragmatic nature of the issue discussed, DES potential as an UV filter, and not as a sunscreen, should have been determined by, at least, reflectance spectroscopy with integrated sphere to in vitro obtain an efficacy profile against UVB and/or UVA radiation (SPF and critical wavelength, for instance), in addition to its functional photostability. Yet, it is suggested to combine traditional UV filters to evaluate the DES potential as a future UV filter candidate in an adequate vehicle.

Antioxidant assay by DPPH seemed to be incompletely presented, since no positive control was described.

Reviewer 1

We thank **Reviewer 1** for their careful reading of our manuscript. We have responded to all comments and concerns raised, and any changes made to the manuscript reflecting these appear in **red** in the revised manuscript and Supplementary Information. Please note, changes in **blue** pertain to changes we have made in response to the comments raised by **Reviewer 2**.

Comments: The paper by Horbury et al., reports the synthesis and evaluation of a new candidate for sunscreen formulations. The photophysical properties of the new compound together with its endocrine disruption potential and antioxidant activity are presented. Overall, the paper has undoubted merits as it tries to provide a comprehensive view of the behavior of the system. However, some parts of the manuscript lack the required level of detail. Clearly, the paper is not presented as a complete study and a more detailed analysis would be needed before the real potential of this molecule could be understood. Nevertheless, some points in the manuscript should be improved before publication.

1. The symmetrical nature of the DES is invoked several times (including the paper title) as responsible for some of the improved properties with respect to ES. Certainly, this is a clever design and the point may well be true. The lack of photoproducts (as the photoisomerization leads to exactly the same compound) is definitively interesting. However, it is not clear if the symmetry along the central double bond is required for this. For instance, an asymmetric compound with, for instance, two different ester moieties, should probably have the same photophysical properties (as it happens with ES and DES). For two similar ester moieties, the UV absorption of both isomers should be very similar. In addition, as the *cis* isomer of ES is genotoxic, this hypothetical asymmetric compound could also provide negative endocrine disruption activity, as it happens with DES. Thus, is it really necessary to have a symmetric compound? If not, DES could be just an example of a set of compounds, based on ES, with improved features in terms of toxicity (but very dependent on substitution) and, maybe, in terms of photostability.

Response: The reviewer raises an interesting point regarding asymmetrical functionalization of the acrylic double bond. This would be an attractive avenue of future research, particularly when trying to tailor the properties of the sunscreen filter, potentially opening a plethora of sunscreen variants. We have therefore altered our discussion to suggest the possibility of using asymmetric ester moieties; however, with the caveat that in doing so, extra experimental considerations may be warranted *i.e.* testing isomeric genotoxicity.

Action: We have added additional text to the discussion:

Page 5, Column 2, Paragraph 3:

Our studies may thus provide a blueprint for tuning molecular functionality, ultimately aiding additional beneficial properties such as enhanced antioxidant potential by modifying the ester group, which has little impact on the photodynamics (cf. **ES** versus **DES**).^{2,3,4,7} **It is worth noting that the ester groups may not necessarily need to be symmetrical *per se*. Asymmetrical substitution, provided the intrinsic properties of the *cis*- and *trans*-isomers are commensurate, *i.e.* photophysical and photochemical properties, antioxidant capacity and endocrine disruption activity, could be promising sources of additional sunscreen filters. However, such an asymmetric substitution may require more sophisticated (*i.e.* expensive) synthetic procedures and additional considerations when testing for suitability.** Our demonstration of customized photoprotection has also much broader ramifications;

not only can it be applied to new materials for human photoprotection, but also towards developing new materials for plastics and resins, given their exposure to UV radiation.

2. Figure 1 includes a very schematic diagram of the processes involved in the photoisomerization of ES. Based on the results included in the paper, the case of DES is more complex: the additional dark excited state species suggested imply the presence of, at least, two different excited states. As stated by the authors, a deep study of the photochemistry of the compound is out of the scope of the paper. I agree with this statement. However, the lack of more detailed information on the mechanism makes risky to draw the conclusions offered by the authors.

The manuscript is definitively interesting and some of the results are relevant enough to be published in Nature Communications. However, I feel that the present version is too preliminary and the above-mentioned issues should be improved before publication.

Response: We apologise that we did not provide a more detailed explanation of the mechanism. This was, impart, due to the limitation we encountered with fitting the transient absorption spectra, owing to the poor signal-to-noise from the **DES** VC/AB data. To remedy this and provide a more complete set of conclusions regarding the mechanism, we have redesigned our transient electronic absorption spectroscopy (TEAS) setup; this has delayed our response to these comments. Specifically, the addition of off-axis parabolic mirrors post sample to better-collect the transmitted white-light probe (which previously suffered from scatter from the textured semi-transparent skin, Vitro-Corneum, surface). With the substantially improved signal-to-noise in our TEAS data, this has allowed us to globally fit the data (something queried by Reviewer 2 as well) with a more refined model. This, in turn, has allowed us to gain a more detailed insight into the photochemistry of **DES** in the various (skin/solvent) environments. These new insights are reflected in the substantial changes to the discussion of the **DES** photochemistry. Some of the actions we have taken in the revised manuscript, in response to this **Point 2** (above), are presented below. We add however, that further actions linked to **Point 2** (above) are also discussed in our response to **Reviewer 2**.

Action: Added additional text in the discussion:

Page 4, Column 2, Paragraph 1:

We shall now focus on the photochemistry responsible for the apparent **DES** photostability.

To recover the photodynamics observed in the TAS, we performed a sequential global fit across the entire spectral window of our probe. This resulted in four dynamical processes, described by rate constants $k_{1,2,3,4}$, being recovered; we note that k_5 is used to describe the long-lived ground state bleach, which does not recover within the time-window of our experiment (except for the cyclohexane data, in which all spectral features return to baseline). We recognised that there are multiple sources we can attribute this incomplete ground state bleach recovery including a potential molecular photoproduct, or a trapped excited state population such as a triplet state (note an absorption feature and no ground state bleach is observed for **DES** in ethanol, see Supplementary Fig. S3). Triplet states have been seen previously for sinapate and cinnamate esters in the gas-phase, with their population being mediated by an $^1n\pi^*$ state.^{7,24,25} These could be a potential source of the small depletion in our steady-state irradiation studies (see Fig. 3 and Fig. S7), qualitatively in line with the minor contribution of EADS for k_5 (see Fig. 2b).

From inspection of the EADS linked to $k_{1,2,3}$ (except for **DES** in cyclohexane where k_2 is absent, we return to this below) these appear to have similar features to EADS seen in related sinapate esters in polar solvents.^{3,4} Therefore, we believe that the EADS for **DES** describe similar dynamical processes, whereby: k_1 represents a prompt geometry relaxation out of the Franck-Condon region; k_2 describes further evolution along the excited state, including vibrational cooling and solvent rearrangement;

and k_3 is the rate-constant for photoisomerization, repopulating the electronic ground state. Under this description $k_1 = k_{\text{geo}}$, $k_2 = k_{\text{evo}}$ and $k_3 = k_{\text{iso}}$ (see Fig. 1). Importantly, we note that due to the symmetrical nature of **DES**, it is not possible to determine if the photoisomerization occurs completely or is an aborted photoisomerization.²⁶

Interestingly however, additional spectral and dynamical (through the returned k_4) features are observed in the TEAS measurements of **DES** when compared to related sinapate esters (e.g. **ES**) and cinnamates.^{2,4,5,7,15,19} These features pertain to the ground state bleach and an excited state absorption at 380 nm. We believe that these features are likely due to an electronic excited state, the front-runner being most likely a $^1\pi^*$ state, previously implicated to play a role in the photodynamics of cinnamates.^{25,26,27,28} The decay of the absorption at 380 nm corresponds to an almost complete recovery of the ground state bleach. Importantly, whether this additional decay pathway happens following bifurcation of the excited state population or subsequent to photoisomerization is unknown.

Additional Reference

24. Rodrigues NDN, *et al.* Towards elucidating the photochemistry of the sunscreen filter ethyl ferulate using time-resolved gas-phase spectroscopy. *Farad Discuss*, **194**, 709-729 (2016)

25. Kinoshita S-n, *et al.* Direct observation of the doorway $^1\pi^*$ state of methylcinnamate and hydrogen-bonding effects on the photochemistry of cinnamate-based sunscreens. *Phys Chem Chem Phys* (2019).

26. Sampedro Ruiz D, Cembran A, Garavelli M, Olivucci M, Fuß W. Structure of the Conical Intersections Driving the cis-trans Photoisomerization of Conjugated Molecules. *Photochem Photobiol* **76**, 622-633 (2002).

Page 5, Column 2, Paragraph 4:

The TEAS setup used to observe the photochemistry of **DES** has been described in detail previously,^{31,32} however, information specific to the present experiments is provided herein, particular regarding the new optical setup for collecting the probe beam (*vide infra*). Samples of **DES** were made to a concentration of 1 mM in C12–15 alkyl benzoate (Lubrizol), ethanol (absolute, VWR) and cyclohexane (99.99%, VWR). For **DES** on VITRO-CORNEUM® (IMS Inc., VC) a **DES** sample at a concentration of 10 mM in C12–15 alkyl benzoate was applied to VITRO-CORNEUM®. The fs pump pulses were generated by an optical parametric amplifier (TOPAS-C, Spectra-Physics) with a fluence of 200 – 800 $\mu\text{J}\cdot\text{cm}^{-2}$. The probe pulse was a broadband white light supercontinuum generated in a vertically translated CaF_2 window, providing a probe spectral window of 345 – 735 nm. The pump-probe time delay (Δt) was varied by adjusting the optical delay of the probe pulse, the maximum obtainable Δt was 2 nanoseconds (ns). The probe beam is collected after passing through the sample, by a pair of 2" diameter, UV-enhanced aluminium, off-axis-parabolic mirrors with a focal length of 100 mm; these mirrors have been incorporated into the setup to compensate for the additional scatter induced by VC. The beam then passes through a CaF_2 lens to collimate the probe, before being focussed into a fibre coupled spectrometer (Avantes, AvaSpec-ULS1650F) by a second CaF_2 lens. Changes in the optical density (ΔOD) of the samples were calculated from transmitted probe intensities. The sample delivery system was a flow-through cell (Demountable Liquid Cell by Harrick Scientific Products Inc.) consisting of two CaF_2 windows; the windows were spaced 100 μm apart. The sample was circulated using a diaphragm pump (SIMDOS, KNF) and replenished from a 25 mL reservoir to provide each pulse-pair with fresh sample. For VC/AB the sample thickness was measured to be 120 μm and was mounted on the front of a CaF_2 window.

Additional suggestions by **Reviewer 1**:

3. The synthesis of DES is quite straightforward. The claims that previous preparations use toxic compounds and require energy-consuming heating are strictly true, but probably imply unneeded overselling. In this preparation, piperidine (toxic, according to the authors) is also used. The use of microwave and column chromatography is not industrially efficient.

Response: We agree with the reviewer that microwave and column chromatography may not be industrially efficient. Therefore, an additional synthetic pathway was developed (post submission of this work), allowing for multigram production, while at the same time eliminating use of piperidine. In addition, we were able to replace the column chromatography purification with simple precipitation. We believe that this additional synthetic route addresses the concerns raised by this reviewer.

Action: Added additional text to the manuscript:

Page 2, Column 1, Paragraph 2:

(a) developing **two** greener synthetic **procedures** to produce **DES**;

Page 2, Column 1, Paragraph 3:

DES can be readily obtained through the Knoevenagel condensation of syringaldehyde and diethyl malonate. Current synthetic procedures that have been reported in the literature for this condensation are not only quite hazardous, as they use toxic reagents/solvents such as piperidine and benzene, but they also require energy consuming-conventional heating. As we dedicate ourselves to the development of sustainable chemical pathways, our first goal was to design a greener synthesis of **DES**. By using microwave heating instead of conventional heating, we were able to reduce energy consumption, eliminate benzene while reducing the reaction time from 7.5 hours to 30 minutes. **Although this procedure brought significant improvements, it still required piperidine; moreover, one could also question the relevancy of microwaves at the industrial scale. A proline-mediated Knoevenagel-Doebner condensation in ethanol¹⁶ under conventional heating, recently developed in our group, was successfully implemented to the synthesis of **DES** at the multigram scale allowing for full replacement of piperidine. Finally, whatever the synthetic procedure used (*i.e.*, microwave-assisted or proline-mediated condensation), we also succeeded in replacing column chromatography purification by a simple precipitation. Ultimately, this leads to a more sustainable and environmentally friendly synthetic route of **DES**.**

Page 6, Column 1, Paragraph 3:

Synthesis of DESs

DES was synthesized in one step using either a micro-wave-assisted or a proline-mediated Knoevenagel condensation between lignin-derived syringaldehyde and diethyl malonate (Scheme 1).³³

Scheme 1. DES synthesis through Knoevenagel condensation

Procedure 1:

Syringaldehyde (4 mmol, 728 mg) and diethylmalonate (13 mmol, 2 mL) were mixed together. Piperidine (2 mmol, 200 μ L) was then added to the reaction mixture, the tube sealed and placed into a Monowave 400 microwave system. Constant power (50 W) was applied until reaching a temperature of 100°C which was then maintained for 30 additional minutes.

The reaction mixture was purified by either, **Purification 1**: flash chromatography (cyclohexane/ethyl acetate 8/2). Fraction containing the wanted product was evaporated under vacuum to give pure **DES** (1.1 g, 85%); or **Purification 2**: The reaction mixture was cooled down to room temperature and added dropwise to a 1N HCl aqueous solution (50 mL) at 0°C. The resulting precipitate was filtered and washed with cold water resulting in pure **DES** (1.05 g, 80%).

Procedure 2:

Syringaldehyde (4 mmol, 728 mg) and diethylmalonate (13 mmol, 2 mL) were mixed together. Proline (2 mmol, 235 mg) was then added and the reaction mixture was refluxed overnight. The reaction mixture was cooled down to room temperature and added dropwise to a 1N HCl aqueous solution (50 mL) at 0°C. The resulting precipitate was filtered and washed with cold water to afford pure **DES** (1.04 g, 80%).

Additional reference:

16. Peyrot Cd, Peru AIA, Mouterde LM, Allais F. Proline-Mediated Knoevenagel–Doebner Condensation in Ethanol: A Sustainable Access to p-Hydroxycinnamic Acids. *ACS Sustainable Chem Eng*, **7**, 9422-9427, (2019).

4. Different solvents and different concentrations are used throughout the paper. This makes difficult to compare the data from one experiment to another. For the most concentrated experiment (TAS in AB, Figure 2, 10 mM), this should result in a 3.2% solution which in turn would be far lower in a real formulation (too low for a significant photoprotection, probably).

Response: We apologise for the lack of clarity regarding the different concentrations used for the TEAS measurements. We mentioned in the original manuscript that the 10 mM solution was used to coat VC, which upon being absorbed into VC, effectively diluted the concentration of **DES** due to the physical volume of VC itself. While, we do not know the exact concentration, we do estimate it to be \sim 1.5 mM based on the signal strength observed in the TAS. Therefore, to make it easier to compare the data, we have normalized the TAS Δ OD in the false colourmaps (see revised Fig. 2 and Fig. S2–4 below). Furthermore, at 10 mM, the percentage of **DES** to C12 – 15 Alkyl benzoate (emollient used in this work) is 0.34% by weight. While this is significantly lower than what would be used in a commercial sunscreen formula (octocrylene for example has an upper 10% by weight as defined by EU Standards), increasing this concentration further would prohibit us carrying out our TEAS measurements due to excessive absorption of both our pump pulse and sections of our white light probe.

Action: Normalized TAS ΔOD in the false colourmaps, both in main paper and Supplementary Information.

Fig. 2. a) TAS of **DES** in VC/AB photoexcited at 335 nm, shown as a false colourmap, with the intensity scale representing a change in **normalized optical density (ΔOD)**. The time-delay is plotted linearly from -0.5 to 10 ps then as a log scale from 10 to 100 ps. b) Evolution associated difference spectra from the sequential global fit of the TAS of **DES** in VC/AB photoexcited at 335 nm. c) Selected TAS at specific Δt highlighting the absorption at 380 nm (8 ps, green) and incomplete ground state bleach recovery (2 ns, orange).

Fig. S2. a) TAS of **DES** in AB photoexcited at 335 nm, shown as a false colourmap, with the intensity scale representing a change in **normalized optical density (ΔOD)**. The time-delay is plotted linearly from -0.5 to 10 ps then as a log scale from 10 to 100 ps. b) Evolution associated difference spectra from the sequential global fit of the TAS of **DES** in AB photoexcited at 335 nm. c) Selected TAS at specific Δt highlighting the absorption at 380 nm (8 ps, green) and incomplete ground state bleach recovery (2 ns, orange).

Fig. S3. a) TAS of **DES** in EtOH photoexcited at 336 nm, shown as a false colourmap, with the intensity scale representing a change in **normalized optical density (ΔOD)**. The time-delay is plotted linearly from -0.5 to 10 ps then as a log scale from 10 to 100 ps. b) Evolution associated difference spectra from the sequential global fit of the TAS of **DES** in EtOH photoexcited at 336 nm. c) Selected TAS at specific Δt highlighting the absorption at 380 nm (8 ps, green) and evidence of a long-lived photoproduct (2 ns, orange).

Fig. S4. a) TAS of **DES** in cyclohexane photoexcited at 325 nm, shown as a false colourmap, with the intensity scale representing a change in **normalized optical density (ΔOD)**. The time-delay is plotted linearly from -0.5 to 10 ps then as a log scale from 10 to 100 ps. b) Evolution associated difference spectra from the sequential global fit of the TAS of **DES** in cyclohexane photoexcited at 325 nm. c) Selected TAS at specific Δt highlighting the absorption at 380 nm (8 ps, green) and lack of a photoproduct (2 ns, orange).

5. The discussion on the effects of solvent polarity is confusing. The lack of influence of the viscosity on k_2 (k_{iso}) is linked by the authors to a low amplitude movement (hula-twist?). However, data available could also be consistent with an aborted photoisomerization (and, thus, with a small nuclear movement).

Response: The reviewer raises an interesting point regarding the possibility of an aborted photoisomerization. As such, we have added additional text acknowledging the possibility of an aborted photoisomerization. We also believe that we were over-ambitious with our reading into the effects of viscosity from the data presented in the original manuscript, especially in light of the improvements to our signal-to-noise of our present data. We have therefore elected to remove the discussion of the viscosity data from the revised manuscript and Supplementary Information.

Action: Altered text in the manuscript:

Page 4, Column 2, Paragraph 3:

We note that due to the symmetrical nature of DES it is not possible to determine if the photoisomerization occurs completely or is an aborted photoisomerization.²⁶

Final Response: We thank this reviewer once again for the points they raise. We hope our revisions implemented in both the manuscript and Supplementary Information address these points.

Reviewer 2

We once again thank **reviewer 2** for their careful reading of our manuscript. We have responded to all comments and concerns raised, and any changes made to the manuscript reflecting these appear in **blue** in the revised manuscript and Supplementary Information. Please note, changes in **red** pertain to changes we have made in response to the comments raised by **Reviewer 1**.

Comments: The authors of this manuscript presented transient absorption data on a diethyl sinapate (denoted DES herein), a sinapate ester derivative, as well as other data pertaining to the feasibility of utilising this molecule as an active ingredient in commercial sunscreen such as long-term photostability, endocrine activity and the added benefit of antioxidant activity. Adding an extra ester to make indistinguishable trans- and cis-form (in order to solve the toxicity issue of the cis-form) and to red-shift the absorption spectrum to cover more UVA is clever, and transient absorption experiments performed on VC, a stratum corneum mimic, is novel. However, the authors did not sufficiently motivate how the transient absorption data, the major result of this manuscript which much of the discussion is centred on, helped establishing that DES is a suitable candidate for sunscreen. Furthermore, the authors' interpretation of the transient absorption data is not as convincing as one would hope to see in a publication like Nature Communications (see below). However, given the importance of the subject which will be of interest to a broad range of readers, we recommend a major revision which should address the major and minor concerns listed below, followed by a future review.

1. The subtle differences between the transient absorption data of DES in AB/VC (Figure 2) and those in solutions (AB, ethanol and cyclohexane; Figures S2 – S4), e.g., that the very long-lived state is only observed in the former, and that the intensity of the stimulated emission is different in different environment, are interesting. We recommend that the authors centred the discussion on these differences, which will help improve the novelty of this work. Specifically, what significant insight can experiment in VC offer, compared to the standard experiments performed in solutions?

Response: We thank the reviewer for their suggestions. As discussed in our response to **Reviewer 1** (see response and action to **Point 2**), the new data we have acquired with improved signal-to-noise has enabled us to globally fit the entire TAS and re-evaluate our kinetics (something raised by this reviewer in their **Point 4** below). In doing so, we have untangled subtle features within the TAS of **DES** in different solvent environments. We have thus restructured considerable aspects of our discussion, focusing on the new insight we glean from measurements on our skin model (VC) and how this differs from the standard experiments performed in solution.

Action: Added additional text to the discussion:

Page 4, Column 2, Paragraph 5:

While the overall picture of **DES** photochemistry, mainly the photoisomerization (or aborted photoisomerization) is similar in *all* solvents and constitutes the main finding (from a dynamics viewpoint) of the present work, differences in these dynamics, reflected in the associated rate-constants (see Table 1 and Supplementary Table 1), do exist and warrant discussion. We also choose to focus our discussion on the differences between **DES** in VC/AB compared to **DES** in various solvent environments and, where appropriate, the insights we draw into the intrinsic properties of photoexcited **DES** when mounted on a skin mimic.

First, comparing **DES** in VC/AB and **DES** in AB, the major difference is the almost three-fold increase in k_4 for **DES** in AB compared to **DES** in VC/AB. The reason for this could rest in population trapped in this excited state, experiencing a greater barrier towards ground state recovery for **DES** in VC/AB. Second, comparing **DES** in VC/AB with **DES** in cyclohexane, the six-fold increase in k_4 for **DES** in cyclohexane may also be reconciled by relative barrier heights. Interestingly for **DES** in cyclohexane, there is complete ground state bleach recovery. Understandably, a decreased residence time in this state for **DES** in cyclohexane could also explain the apparent ground state bleach recovery, given there is less opportunity for competing pathways. The (positive) knock-on effects of this could (tentatively) explain our steady-state irradiation data of **DES** in cyclohexane, which show the smallest amount of depletion following prolonged irradiation.

Like **DES** in cyclohexane, the TAS at 2 ns for **DES** in ethanol has no apparent ground state bleach, however the presence of a new absorption feature at ~ 350 nm is likely the cause for the absence of the ground state bleach. This peak appears to grow in as the absorption at 380 nm and the ground state bleach recover. We believe that this feature is due to the presence of the phenoxyl radical generated *via* a step-wise two-photon ionization, as seen in numerous previous studies in related cinnamates and sinapates.^{2,20,21} However, this absorption feature in the TAS is very small, and hence we are unable to confirm its two-photon (pump) dependency through TAS. We add here that the presence of the phenoxyl radical in the present measurements is an artefact of the ultrafast spectroscopic measurements; its two-photon dependence makes it highly unlikely to occur in nature. Unfortunately, the presence of this peak has also hindered our ability to accurately fit the TAS using the sequential global fitting model, thus we were only able to accurately extract k_1 , k_2 and k_3 , as the fit significantly underestimates k_4 for the decay of the absorption at 380 nm and ground state bleach recovery.

Page 5, Column 2, Paragraph 2:

Moreover, the **overall** photodynamics measured for **DES** in an emollient used in commercial sunscreen formulas are consistent when deposited on a synthetic skin mimic. It also demonstrates that whilst the dynamics are mildly dependent on **DES** environment, it highlights the need of ‘as close to a true environment’ real-world settings for these measurements.

2. Similarly, the authors should comment on what information transient absorption can provide while other techniques cannot.

Response: We thank the reviewer for their suggestion. We have added additional text to the manuscript to clarify why we have used transient electronic absorption spectroscopy (TEAS). We add here that there are other ultrafast spectroscopy techniques that utilise different probe methods, such as fluorescence upconversion and femtosecond stimulated Raman spectroscopy; these can provide differential information. However, TEAS has proven to be highly adept at tracking photoisomerization, of cinnamates and sinapate esters, and given its availability in our laboratory, it was our technique of choice. We have highlighted this in the revised manuscript.

Action: Added additional text:

Page 1, Column 2, Paragraph 4:

The combination of time-resolved and steady-state spectroscopies, enables one to link the ultrafast with the ultraslow dynamics, providing unprecedented insight into how photophysical processes involved at the very early stages of the photon-molecule interaction, influence the longer-term photostability. Furthermore, TEAS has proven to be a powerful tool for observing the photoisomerization of sinapate esters, particularly in identifying the formation of any photoproducts.^{2,3,4,5,7,15} TEAS measurements were taken of DES blended with a commercial sunscreen emollient, C12-15 alkyl-benzoate (AB) deposited on a synthetic skin mimic, VITRO-CORNEUM® (VC).

3. The description and discussion of the transient absorption results (text in the bottom of the right column of page 2 and the top of the left column of page 3) is confusing. The authors stated that the excited state absorption and the ground state bleach “decay at different rates” (once here and once in the beginning of Discussion), but as the reviewer understands, the fastest three decay components ($k_1 - k_3$) are identical between the two features, an additional long-lived component (k_4) is required to fit the ground state bleach trace. Furthermore, the fact that four decay components were observed in the ground state bleach would argue against a sequential decay mechanism, which was used by the authors to model their transient absorption data. Although the multiexponential nature of the ground state bleach recovery does not totally rule out sequential decay, especially in the presence of overlapping absorption signals, the author should at least comment on the possibility of a parallel mechanism.

4. It is a strange approach where the authors exclude the negative transient absorption signal due to ground state bleach in their global analysis. There is no reason why global fitting would fail when adding an additional decay component (from three to four exponentials).

Response: Since **Points 3** and **4** address similar concerns, we have chosen to address these together. Firstly, we apologise for the lack of clarity with the fitting model presented in the paper. We initially tried several different approaches to fit our data. However, due to what appeared as a difference in decay between the long-lived excited state absorption and ground state bleach, each model we chose that incorporated both features, resulted in poor fits. We had suspected this was due to the poor signal-to-noise in our measurements. Having now had the opportunity to retake these measurements following improvements to our setup (and we thank these reviewers for this opportunity), we have confirmed our original suspicions. As described in our response to **Reviewer 1 (Point 2)**, we have repeated the TEAS measurements, implementing experimental upgrades to improve our signal-to-noise. The new TEAS data accrued has enabled us to carry out accurate sequential global fits across the full spectral window, which includes the ground state bleach, as queried by this reviewer.

Regarding the use of a sequential model, we agree with the reviewer that the multi-exponential nature of the ground state bleach recovery is suggestive of a non-sequential model. For example, as we mentioned in our response to **Reviewer 1 (Point 2)**, we are unsure whether population of the $^1n\pi^*$

state (which subsequently leads to ground state bleach recovery) happens following bifurcation of the excited state population or subsequent to photoisomerization. Likewise, the ground state bleach is also heavily masked with the excited state absorption (e.g. phenoxyl radical absorption; see **Point 1** above). Therefore, whilst a parallel model is undeniably more appropriate, the flipside is that it is very difficult to pin down and hence our choice of a sequential model.

Action: Repeated TEAS measurements with upgraded experimental setup.

See Reviewer 1 **Point 2** for text regarding new experimental setup

Page 3, Column 1, Paragraph 2:

Whilst, it appears that these features decay completely back to the baseline, closer examination of the TAS at $\Delta t = 2$ ns, shown in Fig. 2c, shows that a very small amount of the ground state bleach is still present. We note that we do not see evidence of vibrational cooling of the electronic ground state, as previously seen for ES.⁴ This is due to the presence of the absorption feature at 380 nm masking the spectral signature associated with vibrational cooling. The absence of vibrational cooling has been seen in several other sinapates and cinnamates.^{2,3,21}

Altered text regarding fitting:

Page 3, Column 1, Paragraph 3:

To recover the kinetic parameters from the TAS presented (see Fig. 2a and Supplementary Fig. S2-4), we carried out a sequential ($A \xrightarrow{k_1} B \dots \xrightarrow{k_n} n$) global fit, across the entire spectral region of our probe, using the software package Glotaran.^{22,23} The rate-constants (k_n) for DES in VC/AB returned from the sequential global fit are shown in Table 1 (the corresponding rate-constants for DES in other solvents is shown in Supplementary Table S1), while the evolution associated difference spectra (EADS) are shown in Fig. 2b; the residuals are shown in Supplementary Fig. S6.

Updated Table 1:

Table 1. Rate-constants (k_n) resulting from the sequential global fit of the TAS of DES in VC/AB shown in Fig. 2a. The errors are quoted to 2σ . Rate-constants for DES in AB, ethanol and cyclohexane can be found in Supplementary Table 1.

	k_1 / s^{-1} ($\times 10^{12}$)	k_2 / s^{-1} ($\times 10^{12}$)	k_3 / s^{-1} ($\times 10^{11}$)	k_4 / s^{-1} ($\times 10^{10}$)	k_5 / s^{-1} ($\times 10^8$)
VC/AB	7 ± 2	3.0 ± 0.3	4.24 ± 0.07	1.02 ± 0.06	$\ll 5$

Altered Figure 2: See response to **Reviewer 1, Point 4**.

Added additional text and figures to Supplementary Information: See response to **Reviewer 1, Point 4**.

5. The authors stated that the very long-lived excited state observed in AB/VC “is most likely an $n\pi^*$ state”, but this assignment is not substantiated. This state, with decay rate constant of k_4 , showed large variation in decay rate in different environment with different viscosities (Table S1). Based on this observation, the authors later stated that it “has large amplitude nuclear motion involved during the dynamics” (top of the right column on page 4). Are these two assignments mutually exclusive?

Response: As discussed in our response to **Reviewer 1 (Point 5)** we have elected to remove the viscosity data from the present work. That being said, to address this reviewers’ pertinent question

regarding whether the $^1n\pi^*$ assignment is mutually exclusive with a large amplitude motion, there is no such requirement. A molecule undergoing a large amplitude motion can experience friction from the surrounding solvent in any number of electronic states, not just $^1n\pi^*$ states. A highly appropriate example being azobenzene, which undergoes photoisomerization on both the $^1\pi\pi^*$ and $^1n\pi^*$ (Hermann R. Photoisomerization of azobenzenes. *Photochem Photophys* **2**, 119-141 (1990)).

Action: Based on the removal of our viscosity data in the revised manuscript, no further action has been taken.

6. Related to comment #5 above: the authors tried to draw attention to that the decay rates change as the solvent viscosity changes, but the dielectric constants of the solvents are also different. Can the observed difference in rate due to dielectric environment instead of viscosity?

Response: See **Point 5** above. Additionally, we add here that (in complete agreement with this comment) for a comparison to be more robust, we would need to select solvents with similar dielectric constants but varying coefficient of viscosities.

Action: Removed text from the manuscript.

7. Page 2, right column, first paragraph below Figure 2 caption: it is very difficult to see those three features from Figure 2a.

Response: We agree with the reviewer that it is hard to see all three spectral features, particularly the stimulated emission for **DES** in VC/AB as it is very weak. We have altered the text referring to these features to describe features that are more apparent.

Action: Altered text:

Page 2, Column 2, Paragraph 3:

The geometry relaxation reveals three distinct spectral features that consist of: (i) a ground state bleach (~350 nm) corresponding to where the **DES** electronic ground state absorbs; (ii) a strong excited state absorption (~380 nm) and (iii) a second weaker excited state absorption (~540 nm).

8. Page 2, Figure 2 (and Figures S2 – S4): it is very difficult to compare the decay traces in panels b and c because they are plotted on different time scales.

Response: We have significantly altered these Figures, which no longer include decay traces. See our response to **Reviewer 1, Point 4**.

Action: No further action taken than that provided in response to **Reviewer 1, Point 4**.

9. Page 3, the first line of the left column, it is unclear how the authors rule out radical.

Response: We apologise for the lack of clarity as to why radical formation was ruled out. The radical is ruled out as the feature that could be attributed to it, the absorption at 380 nm, decays as the time-delay is increased; if the radical were responsible for these features, it would persist beyond our maximum time-delay (2 ns) as previously seen for the sinapate esters. Whilst, we would normally perform a detailed pump-power dependency on the feature to rule out a two-photon process, the signal strength of the absorption at longer times is too weak for us to perform with any accuracy. We do however note that in ethanol it is possible that the radical is responsible for the long-lived positive absorption at ~350 nm. The presence of the radical in polar solvents compared to more non-polar solvents has previously been seen in cinnamates and sinapates.

Action: Altered text to the manuscript:

Page 3, Column 1, Paragraph 2:

The decay of these features' rules out the formation of the radical as it would be expected to persist beyond the maximum Δt .^{2,4,6,7,19,20}

10. Page 3, Figure 3 and associated text, the authors drew attention to the fact that a 3.3% decrease of DES is observed after two hours of UVA irradiation, and compare to the easily degradable trans-ES. However, how is this number compared to the common sunscreen molecules that are used in commercial sunscreen? Although the data seem to indicate that a small fraction of DES is degraded, but is there any harmful photoproduct formed (e.g., degrade to ES which the cis form is toxic)? Is this something that the authors can characterise?

Response: The reviewer raises an interesting point regarding the percentage degradation. There is currently no regulator standard on photodegradation for sunscreen filters, however, studies have been done to determine the level of degradation over time in commercial sunscreen formulas. J. Hojerová *et al.* studied the photostability of a selection of sunscreen formulas and saw anything between a 3% to 53% degradation, though they stated that a 20% degradation seemed to be the norm. (Hojerová J, Medovčíková A, Mikula M, Photoprotective efficacy and photostability of fifteen sunscreen products having the same label SPF subjected to natural sunlight. *Int J Pharm* **408**, 27-38 (2011))

As for an identification of a photoproduct, we were unable to determine the presence of any new structures from our ¹H NMR spectrum taken after irradiation so we are unable to comment on what the degradation product is.

Action: none taken.

11. Why do the authors believe that DES is the “next generation of sunscreen”? What is the advantage of DES compared to the current generation molecules and their blends? For example, what is the advantage of DES compared to zinc oxide which physically blocks UVA?

Response: Whilst current sunscreen formulations are effective at blocking UV radiation from reaching the skin when applied correctly, most of the commonly used sunscreen filters within a blend, including some oxides, have come under considerable scrutiny in recent years. This stems from their human toxicity (*viz.* phototoxicity and photoallergy) and ecotoxicity. In the case of zinc oxide, concerns are centred on the impacts the environment and human health using zinc oxide nano particles. (Sharma V, Anderson D, Dhawan A. Zinc oxide nanoparticles induce oxidative DNA damage and ROS-triggered mitochondria mediated apoptosis in human liver cells (HepG2). *Apoptosis* **17**, 852-70 (2012); Bai W, Zhang Z, Tian W, He X, Ma Y, Zhao Y, Chai Z. Toxicity of zinc oxide nanoparticles to zebrafish embryo: a physicochemical study of toxicity mechanism. *J. Nanoparticle Res* **12**, 1645-54 (2010)) Research and development into nature inspired sunscreen formulations is certainly an attractive avenue, as these nature-derived products may well have reduced human- and eco-toxicity.

Action: We have added additional text to the introduction to add clarity at the need for new sunscreen filters:

Page 1, Column 2, Paragraph 2:

One must also keep in mind that given the growing concern over several other EU and FDA approved sunscreen filters flagged as human-toxic^{10,11} and eco-toxic,^{12,13,14} this inevitably adds further considerations before any new sunscreens agent can be included in a sunscreen formulation.

Final Response: We thank this reviewer once again for the points they raise. We hope our revisions implemented in both the manuscript and Supplementary Information address these points.

Reviewer 3

We once again thank **reviewer 3** for their careful reading of our manuscript. We have responded to all comments and concerns raised, and any changes made to the manuscript and Supplementary Information reflecting these appear in **green** in the revised manuscript.

Comments: Dear corresponding Author,

The manuscript was adequately written, containing original data regarding the potential use of DES as an UV filter.

1. Considering the high pragmatic nature of the issue discussed, DES potential as an UV filter, and not as a sunscreen, should have been determined by, at least, reflectance spectroscopy with integrated sphere to in vitro obtain an efficacy profile against UVB and/or UVA radiation (SPF and critical wavelength, for instance), in addition to its functional photostability. Yet, it is suggested to combine traditional UV filters to evaluate the DES potential as a future UV filter candidate in an adequate vehicle.

Response: We thank this reviewer for the suggestion to add information about the SPF and critical wavelength for **DES**. Regarding acquiring an SPF value for **DES**, this is not possible as **DES** primarily absorbs in the UVA; this is one of the attractive features of this sunscreen filter (which we tried to highlight in the introduction) given the prevalence of UVB sunscreens on the market and notable absence of UVA filters. Therefore, we can only provide a critical wavelength value of **DES**. To this end, we have performed measurements of **DES** in ethanol to determine the critical wavelength and compared this to the critical wavelength of **ES** and avobenzone (a current UVA sunscreen filter) in ethanol.

Action: Performed additional experiments to determine the critical wavelength of **DES**:

Added additional text:

Page 3, Column 2, Paragraph 2:

In addition to the photostability measurements of **DES**, we have also calculated the critical wavelength of **DES** from its UV/visible spectrum in ethanol; critical wavelength is the industrial standard for determining if there is sufficient UVA protection. The value we retrieved was 364 nm (see Supplementary Fig. S8).

Page 4, Column 1, Paragraph 3:

Alongside this increase in photostability, the critical wavelength of **DES** has significantly red-shifted compared to **ES**, *cf.* 364 nm for **DES** compared to 352 nm for **ES**. While this falls slightly short of current UVA filters, *i.e.* avobenzone is 378 nm, it is clearly a step in the right direction (see Supplementary Fig. S8 for details).

Added figure and text to Supplementary Information:

Figure S8: UV/visible spectra of **DES** (red), **ES** (black) and avobenzone (blue, Avo) in ethanol. The calculated critical wavelengths are **DES** = 364 nm, **ES** = 352 nm and Avo = 378 nm; these are marked by the vertical line in the corresponding colour.

The critical wavelength for a sunscreen is defined as the wavelength at which the integrated area underneath the spectral absorbance curve reaches 90% of the total area between 290 and 400 nm.¹ To attain the critical wavelengths of **DES**, **ES** and avobenzone, UV/visible spectra of each compound were taken in ethanol using a UV/visible spectrometer (Cary 60, Agilent Technologies). The area under each absorption curve between 290 and 400 nm was determined using the cumulative trapezoidal method function in MATLAB (R2017b), which is defined mathematically as follows:

$$\int_{290}^{400} f(\lambda) d\lambda \approx \sum_{k=1}^N \frac{f(\lambda_{k-1}) + f(\lambda_k)}{2} \Delta x_k$$

where $\lambda_0 = 290 \text{ nm} < \lambda_1 < \dots < \lambda_{N-1} < \lambda_N = 400 \text{ nm}$, and Δx_k is the interval between each wavelength datapoint. The critical wavelength was then assigned to be the value where 90% of the total area resides under the curve.

Reference

1. Diffey BL, Tanner PR, Matts PJ, Nash JF, In vitro assessment of the broad-spectrum ultraviolet protection of sunscreen products, *J. Am. Acad. Dermatol.*, **6**, 1024-1035 (2000)

2. Antioxidant assay by DPPH seemed to be incompletely presented, since no positive control was described.

Response: We are (respectfully) unsure on what the reviewer means regarding a positive control for the DPPH assay, as the technique does not require one. However, we have added additional DPPH assays on antioxidant molecules that are used in current commercial sunscreen formulas for further comparison.

Action: Added results from additional DPPH assays of several antioxidants currently used in commercial sunscreen formulas:

Added additional text and table to manuscript:

Page 4, Column 1, Paragraph 2:

For ease of comparison with other studies, we have converted this value to a ratio, quoted as [antioxidant]/[DPPH]; this gives a value of 0.86 for **DES**. In addition, we have included DPPH assays on several commercially available antioxidants, presented in Table 2.

Table 2. EC₅₀ (nmol) and [antioxidant]/[DPPH] values for **DES**, **ES** and several commercially available antioxidants

	Irganox 1010	Trolox	BHT	BHA	ES	DES
EC ₅₀ / nmol	6.9	4.0	7.1	3.7	13.7	32.7
[antioxidant]/[DPPH]	0.18	0.11	0.19	0.10	0.36	0.86

Altered text:

Page 5, Column 1, Paragraph 4:

Whilst the DPPH assays demonstrate that **DES** can act as an antioxidant, its activity (0.86) is lower than **ES** (0.36),⁸ as well as antioxidants already used in commercial sunscreen formulas BHT (0.19), BHA (0.10) and α -tocopherol (0.21).³⁰ These antioxidants are only included in sunscreens in small quantities compared to sunscreen filters. Therefore, while the antioxidant potential of **DES** is lower, its concentration will be significantly greater, thus alleviating its low antioxidant activity. We have also included Irganox 1010 (0.18) and Trolox (0.11) used, respectively in the polymer and pharmaceutical industry.

Page 6, Column 2, Paragraph 3:

This procedure has been applied to commercially available antioxidants to provide benchmark values: Irganox 1010 antioxidant used in polymers, Trolox antioxidant used in the pharmaceutical industry, and BHT and BHA antioxidants are used in the cosmetic and food/feed industries.

Final Response: We thank this reviewer once again for the points they raise. We hope our revisions implemented in both the manuscript and Supplementary Information address these points.

Reviewers' comments:

Reviewer #1 (Remarks to the Author):

This revised version of the manuscript by Horbury et al. addresses most of the issues raised in the previous version. However, it remains unclear the nature of the ground state bleach. As stated by the authors, this could be due to a photoproduct or triplet state formation. Beyond the mechanistic interest, the precise details of this species could be relevant for the assessment of the safety of these compounds as sunscreens.

Reviewer #2 (Remarks to the Author):

We appreciate the authors' efforts in redesigning their transient absorption spectrometer, obtaining new and improved data, and rewrote much of the paper, especially the discussion section. We are satisfied in part with the authors' approach in the revised manuscript. However, we still have concerns over the authors' handling and interpretation of the transient absorption (TA) data.

1) Whether it is sensible to fit TA data to many exponential component (in this case, five) is somewhat subjective, but we noticed that the fourth and fifth components have extremely small amplitudes, possibly representing no more than a few percent of the total TA signal. The k5 components look like a flat line at zero in Figure 2, and Figure S2-S4 in the SI. The k4 components at the ground state bleach (GSB) region are also very small (with the exception of the ethanol solution). We therefore question whether adding these two components truly improve the quality of the fit, and whether the authors can confidently extract the rates of these weak and slow components from the TA data by global fitting. In fact, by looking at panel c in Figure 2 and Figures S2-S4, we cannot say that the 2 ns traces are different from the baseline (perhaps with the exception of Figure 2 in the main text, which marginal difference in the GSB region is seen). Yet, there are extensive discussion based on these two components in the revised manuscript, e.g., the blue-highlighted text in the left column of page 5. This runs into the risk of over-interpreting the data. Are these small signals merely an artifact, or background signal from solvent or the skin mimic (VC)? The authors did not present any background of the AB/VC mixture. At any rate, we feel that the blue-highlighted text on page 5 is too speculative.

2) The k1 components presented in Table 1 and Table S1 correspond to decay times of 14 – 140 fs, and the k2 components are also on the subpicosecond time scale. Such ultrafast decay falls within

the instrument response time of a typical TA spectrometer. If the authors believe that these numbers are meaningful, then the time resolution of the instrument, along with methods of producing and characterising the ultrashort pulses, should be disclosed. The details for global fitting, such as whether IRF is included and how the chirp is corrected, should also be carefully discussed.

3) We are confused by the authors' comment on phenoxy radical generated by "step-wise two-photon ionization". "step-wise" two photon ionization is typically observed in experiments using longer pump pulses, such as nanosecond pulses, but not quite possible with femtosecond pulses. We believe the authors means a true two-photon ionization which involves simultaneous absorption of two pump photons, but this point needs clarification from the authors.

4) We echo point #4 raised by Reviewer #1, and are not completely satisfied with the authors' response. At higher concentration, molecular aggregation is possible and this can alter the excited-state dynamics of a molecule. We understand that it is not possible to perform TA experiments on such highly concentrated samples due to technical issues, but the authors should comment on whether there is aggregation (could be obtained from steady-state UV-vis spectroscopy), and discuss whether aggregation could affect the photophysics and photochemistry.

Reviewer #3 (Remarks to the Author):

Dear corresponding Author,

The revised version of the manuscript was considered improved, presenting new interesting data.

As a suggestion, the terminology for your molecule could be adjusted to UV filter for sunscreen use, or simply UV filter. Sunscreen filter is not a common term for this area.

The new data for the DES using the critical wavelength test, that is a broad spectrum parameter, not essentially an UVA test, generated the value of 364 nm, inferior than avobenzene.

Perhaps, to consider DES as an UVA filter candidate, another molecule could be challenged, the oxybenzone (benzophenone-3). The DES maximum absorbance value was between UVB and UVA radiation (Figure S8), indicating potential to develop SPF or to be considered a broad spectrum UV filter.

Reviewer 1

We thank Reviewer 1 for their careful reading of our manuscript. We have responded to all comments and concerns raised, and any changes made to the manuscript reflecting these appear in red in the revised manuscript and Supplementary Information.

Comment: This revised version of the manuscript by Horbury et al. addresses most of the issues raised in the previous version. However, it remains unclear the nature of the ground state bleach. As stated by the authors, this could be due to a photoproduct or triplet state formation. Beyond the mechanistic interest, the precise details of this species could be relevant for the assessment of the safety of these compounds as sunscreens.

Response: As we stated in the manuscript, our symmetrically functionalised sinapate ester may show promise as a nature-inspired sunscreen filter in next generation sunscreen formulations. Before this is realised however, further experiments are necessary that transcend the mechanistic interest. As this reviewer comments, understanding the nature of this potential photoproduct is one such future step. Whilst this is beyond the scope of the present work, we have made a first attempt at addressing this in the present response.

In our efforts to explore this potential photoproduct further, we have carried out high pressure liquid chromatography-high resolution mass spectrometry (HPLC-HRMS using a QToF for the HRMS). In this experiment, we dissolved approximately 3.25 mg of **DES** in 10 mL of ethanol (one of the solvents used in the present measurements). The chromatogram at time = 0 was recorded (see Figure R1, Top). The same solution was then exposed to UV irradiation at 300 nm (the wavelength available on our current HPLC-HRMS setup) for 2 hours. The chromatogram at time = 2hrs was then recorded (see Figure R1, Bottom).

Figure R1: HPLC chromatograms of pre-irradiated (Top; t=0) and post-irradiated (Bottom; t=2hrs) **DES** in ethanol. The irradiation was at 300 nm, with an average power of 8.32 W/m².

As is clear from the data above, and the fact that the HRMS spectra of the sinapate ester peak pre- and post-irradiation are identical (spectra can be provided if required), there are no new peaks emerging from irradiating the sample (**DES** appears at $t = 10.03$ min). Interestingly, the irradiation source was a factor of 20x greater in intensity than the solar incidence at 300 nm. We specifically increased the power to seek out any potential photoproducts with different arrival times (which Figure R1 shows there are none). This finding is in accordance with our NMR irradiation studies (not shown, again spectra can be provided if required) in which irradiation at 336 nm at solar incidence intensities showed very little difference between the pre- and post-irradiated samples. Overall this serves to further support the impressive photostability of **DES**. One point of note is that the peaks at 7.4 and 9.6 are attributed to impurities in the **DES** sample.

Action: None taken.

Reviewer 2

We thank Reviewer 2 for their careful reading of our manuscript. We have responded to all comments and concerns raised, and any changes made to the manuscript reflecting these appear in blue in the revised manuscript and ESI.

We appreciate the authors' efforts in redesigning their transient absorption spectrometer, obtaining new and improved data, and rewrote much of the paper, especially the discussion section. We are satisfied in part with the authors' approach in the revised manuscript. However, we still have concerns over the authors' handling and interpretation of the transient absorption (TA) data.

Comment 1: Whether it is sensible to fit TA data to many exponential component (in this case, five) is somewhat subjective, but we noticed that the fourth and fifth components have extremely small amplitudes, possibly representing no more than a few percent of the total TA signal. The k_5 components look like a flat line at zero in Figure 2, and Figure S2-S4 in the SI. The k_4 components at the ground state bleach (GSB) region are also very small (with the exception of the ethanol solution). We therefore question whether adding these two components truly improve the quality of the fit, and whether the authors can confidently extract the rates of these weak and slow components from the TA data by global fitting. In fact, by looking at panel c in Figure 2 and Figures S2-S4, we cannot say that the 2 ns traces are different from the baseline (perhaps with the exception of Figure 2 in the main text, which marginal difference in the GSB region is seen). Yet, there are extensive discussion based on these two components in the revised manuscript, e.g., the blue-highlighted text in the left column of page 5. This runs into the risk of over-interpreting the data. Are these small signals merely an artifact, or background signal from solvent or the skin mimic (VC)? The authors did not present any background of the AB/VC mixture. At any rate, we feel that the blue-highlighted text on page 5 is too speculative.

Response: The reviewer raises an understandable concern regarding the number of exponential components that we have used within our fits, given the small amplitude of the k_4 and k_5 EADS. Since these are mainly pertinent to our discussion on **DES** in VC/AB and AB we shall focus on them. The spectral evolution within the TAS that gives rise to k_4 EADS is around $0.5 - 1$ m Δ OD in magnitude, well above the signal-to-noise of the TEAS system (0.05 m Δ OD). We are therefore confident that we can accurately fit and extract the k_4 EADS from our TAS.

Regarding k_5 EADS, while the spectral feature in the TAS is close to the limit of what we can measure (*vide supra*), it is consistently present, and unchanging, at all pump-probe time-delays > 500 ps (well beyond k_4 process has completed). Furthermore, it has been present in multiple repeats of the experiments; with new samples, of different synthesis batches, being used each time. We have also ruled-out this feature from being due to solvent-only as TAS of VC/AB and AB alone return to baseline by a pump-probe time delay of 1 ps. We are confident that this is a real spectral feature due to **DES** and not due to a system artefact. Therefore, k_5 is required in the fitting model to accurately reproduce this feature.

We also agree with the reviewer that no structure within the EADS is visible in Fig 2 of the main manuscript and Fig S2 and 3 in the Supplementary Information. However, once the scale is adjusted, one sees the fine structure within the EADS of k_4 and k_5 . We have included revised figures in the Supplementary Information reflecting the above. Just to note that there is no k_5 EADS for cyclohexane as solvent, as discussed in the main manuscript. Consequently, we do not provide a zoomed-in figure therefore for **DES**/cyclohexane.

Action: We have added additional zoomed-in comparison plots of the k_4 and k_5 EADS and the measured TAS (in the appropriate time window) to the Supplementary Information. As well as showing the agreement between the calculated (EADS) and measured (TAS), these serve to demonstrate the evolution of the TAS and the necessity to include both k_4 and k_5 in the model.

Fig. S8 TAS of **DES** in VC/AB overlaid on the corresponding EADS: a) TAS at 20 ps overlaid on the EADS associated with k_4 , and b) TAS at 2 ns overlaid with the EADS associated with k_5 . Note, normalisation is based on the largest amplitude feature.

Action: We have added additional zoomed-in plots of the EADS to the Supplementary Information that show the spectral structures of k_4 and k_5 , and corresponding text in the main manuscript: 'additional zoomed-in plots of the EADS associated with k_4 and k_5 are shown in the Supplementary Fig. S7'.

Fig. S7 Zoomed-in plots of EADS k_4 and k_5 for **DES** in a) VC/AB, b) AB and c) ethanol.

Action: Added solvent-only TAS of VC/AB and AB at 1 ps to the Supplementary Information, and corresponding text in the main manuscript: ‘which is not attributed to the solvent; see Supplementary Fig. S6 for solvent-only TAS’.

Fig. S6 TAS of a) VC/AB and b) AB at 1 ps.

Comment 2: The k_1 components presented in Table 1 and Table S1 correspond to decay times of 14 – 140 fs, and the k_2 components are also on the sub-picosecond time scale. Such ultrafast decay falls within the instrument response time of a typical TA spectrometer. If the authors believe that these numbers are meaningful, then the time resolution of the instrument, along with methods of producing

and characterising the ultrashort pulses, should be disclosed. The details for global fitting, such as whether IRF is included and how the chirp is corrected, should also be carefully discussed.

Response: This is a mistake on our part. We quoted k_1 for **DES** in cyclohexane as $7 \times 10^{13} \text{ s}^{-1}$ which corresponds to a time-constant of 14 fs. This value should be $0.7 \times 10^{13} \text{ s}^{-1}$ corresponding to a revised time-constant of 140 fs (within our instrument response of ~ 80 fs, see below). Our sincerest apologies for this oversight and the confusion this has caused.

Regarding characterization of pulse duration, we have included transient slices of our solvent-only scans showing the time-zero response. These are fitted with a Gaussian function, the instrument response function (IRF) taken as the fullwidth half maximum. The largest rate-constant we quote is half the IRF. The spectral region selected for the transient slice is selected for the most Gaussian-like response, due to the varying nature solution-phase IRFs can take.

Finally, our chirp correction is achieved by including a third-order polynomial dispersion curve, which comes from the solvent only time-zero response, in the global fitting algorithm.

Action: Changed k_1 value for **DES** in cyclohexane quoted in Table S1 to $0.7 \pm 0.2 \text{ s}^{-1} (\times 10^{13})$

Action: Additional text to Supplementary Information regarding fitting procedure: ‘To account for the chirp of our probe pulse, a third order polynomial is included within the fitting algorithm. Additionally, the fit is convoluted with an instrument response function to account for the temporal resolution of our pulses, whose value is taken from Gaussian fits of the solvent-only time zero response (see Supplementary Information Fig. S8 for exemplar fits).’ Added additional text to the main manuscript: ‘further details can be found in the Supplementary Information’.

Action: Added transient slices to the Supplementary Information of solvent-only responses at selected wavelengths for VC/AB, AB, ethanol and cyclohexane, fitted with Gaussians to extract the IRF.

Fig. S9. Selected transients for solvent-only time-zero responses of a) VC/AB (380 nm), b) AB (380 nm), c) ethanol (310 nm) and d) cyclohexane (360 nm). The probe wavelength was chosen which showed the most Gaussian-like response. The solid line represents the fit of the Gaussian function to the experimental data. The extracted fullwidth half maxima are: a) 80 fs, b) 80 fs, c) 80 fs and d) 90 fs. These values are used as our IRFs in the corresponding global fits.

Comment 3: We are confused by the authors' comment on phenoxy radical generated by "step-wise two-photon ionization". "step-wise" two photon ionization is typically observed in experiments using longer pump pulses, such as nanosecond pulses, but not quite possible with femtosecond pulses. We believe the authors means a true two-photon ionization which involves simultaneous absorption of two pump photons, but this point needs clarification from the authors.

Response: We apologize for this confusion. We used the term 'step-wise two-photon ionization' incorrectly. The reviewer is entirely correct that the two-photon ionization is instantaneous (see main manuscript, Ref 20) rather than step-wise. We have therefore removed reference to 'step-wise' in the revised manuscript.

Action: Changed 'step-wise two-photon ionization' to 'instantaneous two-photon ionization' in main manuscript.

Comment 4: We echo point #4 raised by Reviewer #1, and are not completely satisfied with the authors' response. At higher concentration, molecular aggregation is possible and this can alter the excited-state dynamics of a molecule. We understand that it is not possible to perform TA experiments on such highly concentrated samples due to technical issues, but the authors should comment on whether there is aggregation (could be obtained from steady-state UV-vis spectroscopy), and discuss whether aggregation could affect the photophysics and photochemistry.

Response: The reviewer raises an interesting point regarding aggregation. Hydrogen bonded dimers have been observed in sinapic acid, the base molecule of **DES** (see main manuscript Ref 2). However, this was attributed to the carboxylic acid tail, which is now absent in **DES**. Another potential source of aggregation could be intermolecular hydrogen bonding mediated by the OH group, which causes the aggregation seen, for example, in a recent study in catechol;^a however the presence of the flanking OMe groups and the consequent intramolecular hydrogen bonding prevents this aggregation. Thus, the only other source for aggregation which we surmise, would be $\pi\pi$ stacking, which is seen, for example in phenol, and provides a unique spectral signature in its TAS.^b We note that this spectral signature grows in overtime around 600 nm.

To explore such potential stacking, we have performed TEAS measurements at 10 mM in ethanol, the highest concentration we could achieve without excessive absorption of either our pump or probe pulses. In short, the TAS collected at 10 mM versus 1 mM **DES**/ethanol show no discernible differences. This is shown in the normalised TAS presented in Figures R2a and R2b below, following photoexcitation at 336 nm and presented as false colourmaps. Based on this result, we are confident that aggregation is unlikely to occur in **DES**, certainly within the experimental concentrations possible.

Figure R2 Normalized TAS of **DES** in ethanol at a) 10 mM and b) 1 mM. Photoexcitation was at 336 nm.

References:

- Grieco C, Kohl FR, Zhang Y, Natarajan S, Blancafort, Kohler B. Intermolecular hydrogen Bonding Modulates O-H photodissociation in molecular aggregates of a catechol Derivative. *Photochem. Photobiol.* **95**, 2019, 163-175.
- Zhang Y, Oliver TAA, Ashfold MNR, Bradforth SE. Contrasting excited state reaction pathways of phenol and *para*-methylthiophenol in the gas and liquid phases *Faraday Discuss* **157**, 2012

Action: None taken

Reviewer 3:

We thank Reviewer 3 for their careful reading of our manuscript. We have responded to all comments and concerns raised, and any changes made to the manuscript reflecting these appear in green in the revised manuscript and supplementary information.

Dear corresponding Author,

The revised version of the manuscript was considered improved, presenting new interesting data.

Comment 1: As a suggestion, the terminology for your molecule could be adjusted to UV filter for sunscreen use, or simply UV filter. Sunscreen filter is not a common term for this area.

Response: We thank the reviewer for their suggestion and have altered the terminology in the Manuscript to reflect this.

Action: Replaced 'sunscreen filter' with 'UV filter', throughout the manuscript.

Comment 2: The new data for the DES using the critical wavelength test, that is a broad spectrum parameter, not essentially an UVA test, generated the value of 364 nm, inferior than avobenzone. Perhaps, to consider DES as an UVA filter candidate, another molecule could be challenged, the oxybenzone (benzophenone-3). The DES maximum absorbance value was between UVB and UVA radiation (Figure S8), indicating potential to develop SPF or to be considered a broad spectrum UV filter.

Response: One of the main goals of the present research is to develop next generation UVA filters, given the limited selection of such filters on the current market. In our previous response, we used the current standard by the EU and FDA (which is the critical wavelength) to determine the extent of UVA protection.^c

To address this reviewers' comment, and as suggested, we have determined the critical wavelength for oxybenzone. The results are shown in Figure R3 below. We find that the critical wavelength of oxybenzone is 349 nm, whereas the critical wavelength in **DES** is 364 nm. Whilst, as pointed out in our previous response, the critical wavelength of **DES** falls short of avobenzone (378 nm), this finding further supports that, with regards to our original goal, this is a step in the correct direction.

We also add, that as previously mentioned, the absorption profile of **DES** does not sufficiently cover the UV region responsible for erythema, the UVB (280 – 315 nm) region of the electromagnetic spectrum. Consequently, we are unable to determine any significant SPF value. It is clear from the **DES** absorption spectrum that the majority of its absorption, and indeed its λ_{max} , is in the UVA (315 – 400 nm) region compared to the UVB region, therefore making **DES** a potentially promising candidate UVA filter.

Figure R3. Normalized absorbance spectra of oxybenzone (OB, black line), DES (red line) and avobenzone (AB, blue line) in ethanol. Vertical dashed lines denote critical wavelength of associated molecule.

Reference:

- c. Bielfeldt S, *et al.* Multicenter methodology comparison of the FDA and ISO standard for measurement of in vitro UVA protection of sunscreen products. *J. Photochem Photobiol B: Biol* **189**, 2018, 185-192.

Action: None taken

REVIEWERS' COMMENTS:

Reviewer #1 (Remarks to the Author):

The revised version provides new data on the photostability of DES. Although the experimental conditions are clearly different, these results suggest that GSB is not related to photoproduct formation. This would be relevant for any potential application of this compound.

Reviewer #2 (Remarks to the Author):

We are satisfied with the authors response, with the exception of comment 4. The authors performed addition TAS experiments on 1 mM vs 10 mM DES in ethanol, and did not observe differences in spectral features or kinetics. They use this to support their claim that aggregation is unlikely. We note that it is entirely likely that at these concentrations, which are likely to be much lower than the commercial recipe, aggregation may not take place to a significant extent. Furthermore, the authors used the characteristic kinetics and spectral signatures of phenol aggregation as a guide, and argue that because these signals are not observed in DES, aggregation does not occur. We note that different π -stacked systems may have different spectral signatures and rise/decay times, therefore the lack of the signatures for phenol aggregation does not at all rule out DES aggregation. The aggregation problem is beyond the scope of the current study, but we encourage the authors to investigate in their future work. Lastly, we are enthusiastic about this work and the possibility of DES being an UV filter for commercial use. Recent finding that active ingredients of sunscreens can enter human bloodstream calls for much more careful characterisation of a broad range of properties of a sunscreen molecule, such as photostability, toxicity etc. We support the publication of this paper in its current form.

Reviewer #3 (Remarks to the Author):

This manuscript revised version addressed the issues once raised. However, it is suggested an adjustment in the tittle, considering the use of "UVA filter" instead of "sunscreen filter".

This document details our responses to the previous round of comments from our reviewers:

REVIEWERS' COMMENTS:

Comment: *Reviewer #1 (Remarks to the Author):*

The revised version provides new data on the photostability of DES. Although the experimental conditions are clearly different, these results suggest that GSB is not related to photoproduct formation. This would be relevant for any potential application of this compound.

Response and Action: Although no alterations need to be made in light of these comments made by Reviewer 1, we would like to thank them again for their contributions to the manuscript in its current form.

Comment: *Reviewer #2 (Remarks to the Author):*

We are satisfied with the authors response, with the exception of comment 4. The authors performed addition TAS experiments on 1 mM vs 10 mM DES in ethanol, and did not observe differences in spectral features or kinetics. They use this to support their claim that aggregation is unlikely. We note that it is entirely likely that at these concentrations, which are likely to be much lower than the commercial recipe, aggregation may not take place to a significant extent. Furthermore, the authors used the characteristic kinetics and spectral signatures of phenol aggregation as a guide, and argue that because these signals are not observed in DES, aggregation does not occur. We note that different π -stacked systems may have different spectral signatures and rise/decay times, therefore the lack of the signatures for phenol aggregation does not at all rule out DES aggregation. The aggregation problem is beyond the scope of the current study, but we encourage the authors to investigate in their future work. Lastly, we are enthusiastic about this work and the possibility of DES being an UV filter for commercial use. Recent finding that active ingredients of sunscreens can enter human bloodstream calls for much more careful characterisation of a broad range of properties of a sunscreen molecule, such as photostability, toxicity etc. We support the publication of this paper in its current form.

Response: We once again thank Reviewer 2 for their valuable feedback, which has been a consistent feature throughout the review process. More detailed investigations into the aggregation of DES would certainly be an avenue of interest for further research.

Action: The following has been added to the Results section of the manuscript:

“Finally, we note that a more concentrated solution of DES in ethanol (10 mM, the highest possible within our experimental constraints) was tested for evidence of aggregation. There was no evidence (by comparison with the 1 mM counterpart) to suggest that DES aggregates were formed. Further studies would be warranted at higher concentrations, but this is beyond the scope of current experimental capabilities.”

Comment: *Reviewer #3 (Remarks to the Author):*

This manuscript revised version addressed the issues once raised. However, it is suggested an adjustment in the tittle, considering the use of "UVA filter" instead of "sunscreen filter".

Response and Action: We thank Reviewer 3 for highlighting the term “sunscreen filter” still remained in the title, following their previous comment (after the second round of review) which stated that the term “UV filter” would be preferable. The title has been updated accordingly to “Towards Symmetry Driven and Nature Inspired UV Filter Design”.